# QLIP: A Dynamic Quadtree Vision Prior Enhances MLLM Performance Without Retraining

**Kyle R. Chickering, Bangzheng Li, & Muhao Chen**
{krchicke,bzhli,muhchen}@ucdavis.edu

## Abstract

Multimodal Large Language Models (MLLMs) encode images into visual tokens, aligning visual and textual signals within a shared latent space to facilitate cross-modal representation learning. The CLIP model is a widely adopted foundational vision language model whose vision encoder has played a critical role in the development of MLLMs such as LLaVA. However, the CLIP vision encoder suffers from notable limitations including being constrained to only handling fixed input resolutions and a failure to produce separated embeddings for dissimilar images. Replacing the vision encoder of an existing model typically incurs substantial computational costs because such a change often necessitates retraining the entire model pipeline.

In this work, we identify two factors which underlie the limitations of the CLIP vision encoder: **mesoscopic bias** and **interpolation bias**. To address these issues, we propose QLIP, a lightweight adaptation of CLIP that can be seamlessly integrated with existing MLLMs with only a few lines of code and can enhance both coarse-grained and fine-grained visual understanding, without retraining the vision encoder or LLM weights. QLIP is designed around an image quadtree which replaces the standard uniform grid patches with a novel content aware patchification. Our experimental results demonstrate that QLIP improves the general visual question answering accuracy of the LLaVA-1.5 model series across various model sizes—without requiring retraining of the vision encoder or LLM. Notably, QLIP boosts detailed understanding performance on the challenging $V^*$ benchmark by up to 13.6%. Code is available at https://github.com/KyroChi/qlip.

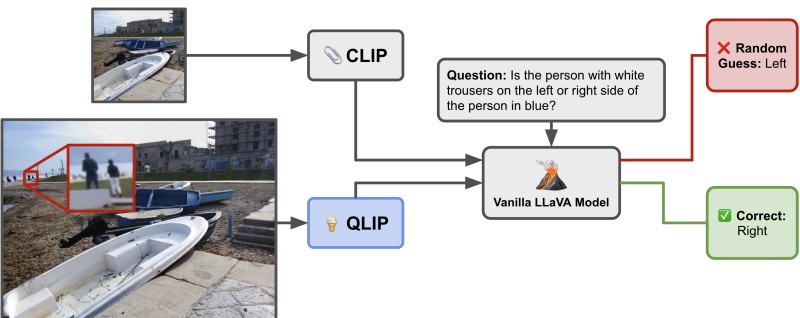

Figure 1: QLIP is a **lightweight adaptation** of CLIP which allows models like LLaVA to perform inference on arbitrarily large images. In our experiments we find that vanilla LLaVA + QLIP gives **+13.6%** accuracy on the challenging $V^*$ benchmark **without retraining the vision encoder or LLM**. The example in the figure above demonstrates an instance where CLIP cannot correctly get the answer because (a) in the cropped version of the image the person in question is not present, and (b) if we use a padded image the person will be too small to provide meaningful signal to the model.

## 1 INTRODUCTION

Multimodal Large Language Models (MLLMs) have shown impressive multi-modal question answering ability, yet recent work has highlighted a deficiency whereby these models struggle to answer questions about fine-grained visual details (Shi et al., 2024; Wu & Xie, 2024). MLLMs, like the popular LLaVA family (Liu et al., 2024a; 2023), use a vision encoder and visual projector to embed visual information into a shared visual-linguistic embedding space before passing these tokens to a downstream LLM. This strategy is not without its flaws. Firstly, authors have observed that a high number of visual input tokens can be removed without significantly affecting performance, indicating redundancy (Hu et al., 2024; Li et al.; Sun et al., 2025). Secondly, it has been shown that models like LLaVA overly rely on information from the vision encoder's `[CLS]` token, which captures global semantics, to answer questions (Zhang et al., 2024).

We posit that this failure on fine-grained VQA tasks is neither a deficiency in the training process of the MLLM nor a deficiency in the representations which can be encoded by the vision encoder. Prior works that have aimed at modifying the vision encoder or projector have implicitly assumed that the failure mode is caused by CLIP itself, but this is only partially true. Li et al. show that the vanilla LLaVA architecture with the CLIP encoder is capable of much better VQA performance, but requires the "correct" tokens to be fed to the language model. Similarly, Li et al. (2024) show that the information from the CLIP encoder is often sufficient for certain vision tasks or VQA, however, the models often do not adequately use the given information. Thus, there remains room to improve MLLM performance by focusing on better use of the available tokens.

We argue that the failures incurred while using the CLIP encoder can be attributed to two specific biases induced by the inductive priors implicitly assumed during CLIP training. **Mesoscopic Bias** occurs because CLIP uses a uniform grid-patchification (UGP) strategy (Dosovitskiy et al., 2020; Radford et al., 2021) and manifests as downstream models implicitly treating uniform grid cells at a specific image scale as the fundamental unit of semantic meaning. **Interpolation Bias** arises as a consequence of CLIP being trained with fixed positional embeddings on fixed-resolution images and prevents CLIP from natively handling high-resolution images.

This work addresses the biases inherent to standard grid-patchification in MLLMs. While newer vision encoders are similarly able to handle high-resolution images, the biases we explore here remain relevant to a broad class of encoders which see continued use. Our method is specifically designed to improve the semantic content of the image tokens for the purposes of VQA.

Previous work has focused on training new vision encoders to replace CLIP (Guo et al., 2024; Liu et al., 2024a; Luo et al., 2024; Shi et al., 2024), but these proposals require re-training the entire MLLM, which is expensive and often not feasible. In this work, we take a minimally invasive approach and carefully reason through the consequences of updated vision priors. This leads us to a *lightweight, content-aware, drop-in* modification to the CLIP encoder which we call QLIP, a portmanteau of "quadtree" and "CLIP".

QLIP empowers CLIP based MLLMs to automatically process arbitrary resolution input images, while adaptively scaling the number of input tokens based on the semantic content of the image. We find that reducing the number of input tokens has beneficial effects beyond faster computation: token reduction can decrease model hallucination and improve fine-grained VQA. To assess both the effectiveness and efficiency of QLIP, we apply our ideas to the LLaVA-1.5 family of MLLMs for VQA. Our method is particularly suited for fine-grained visual tasks like the challenging $V^*$ benchmark (Wu & Xie, 2024). Our method achieves a 13.6% improvement on $V^*$, reduces hallucination rates as measured by the POPE F1 score (Li et al., 2023) by 5.2, and yields improvements across other multi-modal benchmarks including MME (Fu et al., 2023) and RealWorld-QA (xAI Team, 2024).

We accomplish this by using two novel strategies. First, to address the mesoscopic bias we introduce a non-uniform patchification scheme based on image quadtrees (Hunter & Steiglitz, 1979). Our quadtree patchification is *adaptive, tunable, and training-free*, and implicitly treats semantically similar regions of the image as the fundamental unit of semantic meaning instead of UGP. Second, to address the interpolation bias, we train a small MLP network to interpolate the fixed positional CLIP embeddings while maintaining usable positional signals for downstream models.

Our key contributions are as follows:

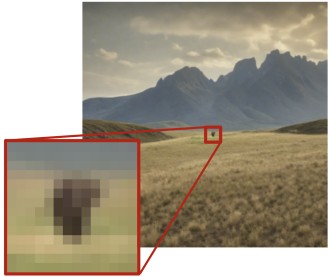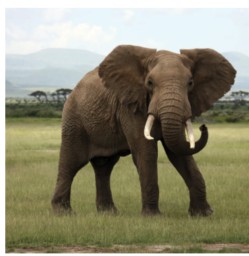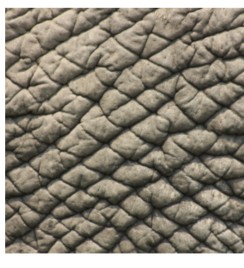

Figure 2: An example of the same semantic feature (`animal:elephant`) at three different spatial scales. These photos could be accompanied by the question `What animal is shown in this photo?` For the leftmost image the elephant fits into a single patch. Without memorization it is unlikely for any classifier to be able to accurately identify the pixelated blob as an elephant instead of, for example, a horse or a buffalo.

1. We identify two fundamental biases in the CLIP vision encoder, i.e. mesoscopic bias and interpolation bias, and propose quantitative measures of both.

2. We introduce QLIP, a lightweight, drop-in modification for CLIP that supports arbitrary image resolutions and adaptively scales the number of the image tokens based on image content. QLIP directly mitigates the aforementioned biases without retraining the vision encoder or LLM.

3. We empirically validate the effectiveness of QLIP by integrating it into the LLaVA model family (Liu et al., 2024a; 2023) and demonstrate substantial performance improvements. Our results are achieved without retraining the vision encoder or LLM. For the challenging $V^*$ benchmark, we achieve a significant improvement of +13.6% accuracy using LLaVA 13B with QLIP, outperforming the previous SoTA CLIP-based LLaVA results by +3.1% (Shi et al., 2024).

## 2 WHY CLIP FAILS AT HIGHER RESOLUTIONS

The CLIP vision encoder is trained at a fixed input resolution using learned absolute positional encodings (Radford et al., 2021). This design introduces two notable and consequential biases. First, because the positional encodings are absolute rather than relative, they do not generalize beyond the spatial grid used during training. Second, the encoder is trained exclusively on fixed-scale images, which biases the encoder towards only recognizing features at a specific mesoscopic spatial scale. For an exaggerated example, consider the elephants in Figure 2. The CLIP encoder is most likely to understand the middle (mesoscopic) image as containing an elephant, rather than the left or right images. This is because during training it is unlikely that the leftmost image would be labeled as having an elephant in it and the rightmost image may be too zoomed in to distinguish it from other concepts. In practice the bias is not this extreme, but as we show in Figure 4 below, changing the image resolution by only a few pixels already substantially decreases the model's ability to recognize the semantic content of an image.

**Quantification of Interpolation Bias** Consider a single image $\mathcal{I}$ rendered at two different resolutions, $R_1 = (H_1, W_1)$ and $R_2 = (H_2, W_2)$. We denote the corresponding resized images as $\mathcal{I}_{R_1}$ and $\mathcal{I}_{R_2}$, respectively. Since both images originate from the same source and contain identical (or nearly identical) semantic content, one would reasonably expect the CLIP [CLS] token embedding to remain invariant or at least approximately constant across these resolutions, especially when $R_1$ and $R_2$ are only slightly different. Under this assumption, the cosine similarity between the corresponding CLIP embeddings, $\mathcal{E}_1 = \mathrm{CLIP}(\mathcal{I}_{R_1})$ and $\mathcal{E}_2 = \mathrm{CLIP}(\mathcal{I}_{R_2})$, serves as a measure of the deviation introduced by resolution changes. To quantify the extent to which positional embeddings contribute to this deviation, we define the interpolation bias as:

$$\mathcal{B}_{\mathrm{Interp}}(\mathcal{I}) := \left\| \nabla_{\mathcal{P}} \mathrm{CS}(\mathcal{E}_1, \mathcal{E}_2) \right\|_2, \tag{1}$$

where $\mathcal{P}$ denotes the additive positional encodings applied to patch embeddings during the CLIP encoding process (Radford et al., 2021) and CS is cosine similarity.

**Quantification of Mesoscopic Bias** Mesoscopic bias is easier to quantify because we can simply remove the positional encodings and look at the cosine similarity of the [CLS] token embeddings at

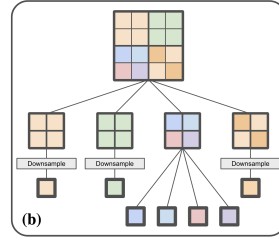

Figure 3: **(a)** An example of the quadtree patchification (QtP) applied to a high-resolution image. QtP uses only 25% of the original number of tokens yet retains a high-degree of semantic information. Photo courtesy of first author. **(b)** A schematic of a $4 \times 4$ patch image being decomposed into 7 leaf patches using a quadtree. Leaves which consist of more than a single patch are downsampled to the patch size.

different image sizes. To this end, consider an image $\mathcal{I}$ with resolution $N \times N$ and then consider the same image rescaled to $336 \times 336$, which we denote $\mathcal{I}_{336}$. Let $\mathcal{E}^z = \mathrm{CLIP}^z(\mathcal{I})$, $\mathcal{E}^z_{336} = \mathrm{CLIP}^z(\mathcal{I}_{336})$ be the respective `[CLS]` embeddings after setting the positional encodings to zero. Then

$$C^z_{N \to 336} := \mathrm{CS}(\mathcal{E}^z, \mathcal{E}^z_{336})$$

captures the degree to which the overall embedding has changed as an effect of the mesoscopic scale of the input images.

## 3    ADDRESSING THE MESOSCOPIC AND INTERPOLATION BIASES

We identify and address two implicit inductive priors underlying the CLIP encoder, noting that these assumptions were likely adopted primarily for engineering practicality.

The first prior is that UGP captures fundamental units of semantics. We address by replacing UGP with a *content-aware quadtree patchification* (QtP). The second prior is that images can be effectively represented by center-cropping and rescaled to a fixed resolution which we address by training a small interpolation network.

### 3.1    VISION QUADTREE MITIGATES THE EFFECTS OF MESOSCOPIC BIAS

Natural images do not contain uniformly distributed information throughout their sub-images. In general, semantic information can continue to be extracted even when large portions of the image are subjected to extreme levels of information degradation at the pixel level (see Figure 3). This is why compression algorithms like JPEG work (Wallace, 1992).

We derive a strategy for adaptively merging adjacent patches in an attempt to increase the quality of the visual signal coming from the vision encoder. This strategy is based on the intuition that many pixels in a given image do not contribute to the representation of the semantic content of the image. We propose using a quadtree (Hunter & Steiglitz, 1979) structure to adaptively select tokens based on some property intrinsic to the sub-images themselves. Quadtrees, as applied in image processing, are hierarchical image representation trees which generalize a binary tree into two dimensions. At the root of the tree is the original image, and at each level we subdivide the image into four, until we reach leaf nodes which represent patches (see Figure 3, **(b)**). We can then prune the tree according to some selection criteria and the resulting leaf-nodes will consist of all sub-images which satisfy some maximal condition. We apply downsampling to the leaf-nodes which are larger than the CLIP encoder's patch size to obtain a sequence of patches that can be fed to the CLIP vision encoder. In theory, semantically irrelevant portions of the image are downsampled back to the mesoscopic scale that CLIP expects, and important tokens which represent a small portion of the visual field are effectively upsampled into the same scale (see Figure 3, **(a)**).

In what follows, we use the following quadtree selection criteria, which can be thought of as the maximum of the average gradient over a patch. Thus, an image $I$ is a leaf-node if it cannot be sub-divided or if

$$\mathcal{D}(I) := \max_{x,y} ( \partial_x I + \partial_y I ) < \alpha, \tag{2}$$

where $\alpha$ is a pre-chosen selection constant. We also test a random selection strategy as an ablation for our selection strategy. More details are contained in Appendix F.

## 3.2 Coordinate-Based MLP Mitigates the Effects of Interpolation Bias

The CLIP vision encoder consists of two mechanisms that work in concert to map information from the pixel space into the embedding space.

Let $\mathcal{P} = \{p_i\}_{i=1}^N$ be a set of patches with coordinates $\mathcal{X} = \{(x_i, y_i)\}_{i=1}^N$, $(x_i, y_i) \in [-1, 1]^2$. The CLIP encoder can be understood as taking $\mathcal{P}$ and $\mathcal{X}$ and producing a sequence of tokens $\mathcal{S} = \{s_i := \boldsymbol{E}(p_i) + \boldsymbol{M}(x_i, y_i)\}_{i=1}^N$ along with a [CLS] token $\boldsymbol{E}_{\text{[CLS]}}(\mathcal{P})$ which captures global information about the image.

CLIP is trained on $336 \times 336$ images decomposed into a series $14 \times 14$ patches using the standard UGP (Dosovitskiy et al., 2020; Radford et al., 2021). There will then be $24 \times 24 = 576$ patches and to each of these patches CLIP associates a positional embedding $\mathcal{E}_{ij} \in \mathbf{R}^{1024}$, where $1 \leqslant i, j \leqslant 24$ respectively index the rows and columns of both $\mathcal{E}$ and the grid of patches. For this patchification we have $\boldsymbol{M}(-1 + \frac{2i}{23}, -1 + \frac{2j}{23}) = \mathcal{E}_{ij}$. We will extend $\boldsymbol{M}$ to the entire square $[-1, 1]^2$ so that we can natively handle images of any resolution and apply our QtP. We choose to train an MLP using our new inductive priors. Choosing an MLP for this task gives us a high-degree of expressivity.

We make the assumption that the [CLS] token should remain invariant when CLIP is applied to a $336 \times 336$ image and the same image at its native resolution[1]. Thus, if $\mathcal{G}$ is the standard UGP associated to the image $I_{336}$ and $\mathcal{P}$ is a patchification associated to the image $I_N$, then we expect that

$$L_{\text{[CLS]}} := \| \boldsymbol{E}_{\text{[cls]}}(\mathcal{G}) - \boldsymbol{E}_{\text{[cls]}}(\mathcal{P}) \|_{L^2} = \text{small}. \tag{3}$$

This provides a target for training the MLP. However, in practice $L_{\text{[CLS]}}$ is insufficient for training since the transformer pooling which generates the [CLS] token means that as long as $\sum_{ij} \mathcal{E}_{ij} = \sum_i \boldsymbol{M}(x_i, y_i)$, then the [CLS] embedding will be constant. Because we are attempting to train a drop-in modification for CLIP and because downstream MLLMs utilize the positional information from CLIP, we must ensure that the MLP positional embeddings match the CLIP positional embeddings on the standard $24 \times 24$ grid. We add a residual $L^1$ error:[2]

$$\mathcal{R}(\boldsymbol{M}, \mathcal{E}) := \frac{1}{576} \sum_{i=1}^{24} \sum_{j=1}^{24} \left| \boldsymbol{M}\left(-1 + \frac{2i}{23}, -1 + \frac{2j}{23}\right) - \mathcal{E}_{ij} \right|. \tag{4}$$

Thus we arrive at a suitable loss function for the MLP training:

$$\text{Loss} = L_{\text{[CLS]}} + \gamma \mathcal{R}, \tag{5}$$

where $\gamma$ is a hyperparameter to balance the relative effects of the two components of the loss. Training is stable and we include additional training details in Appendix C.

## 3.3 Training the Interpolation Network

The coordinate-based MLP must be trained; however, training cost is small relative to MLLM retraining (11 hours on four NVIDIA L40S GPUs). We train the MLP for 100 epochs with the Adam optimizer (Kingma, 2014) on the training split of the Imagenette dataset (Howard, 2019). This dataset is a small subset of Imagenet (Deng et al., 2009) with only 10 classes, and consists of about 10k images. We argue that the choice of dataset does not matter much for the MLP training because the embedding function $\boldsymbol{M}$ is independent of the image content. We train with a batch size of 14, with images kept at either their native resolution or at a resolution with the smallest edge length being 560, whichever is smaller. We kept $\gamma = 1$. For our MLP architecture we use four hidden layers and pass the input features through a Fourier features layer (Tancik et al., 2020) with 48 Fourier features. See Appendix C for a discussion about how we chose model hyperparameters.

---

[1]This assumption is better justified when the native resolution is very close to $336 \times 336$. However we find that generalizing this assumption to arbitrary image resolutions leads to favorable results

[2]We found that $L^1$ loss was better than $L^2$ loss since we aim to get $\mathcal{R}$ to be smaller than $5 \times 10^{-7}$. See Appendix C for more details.

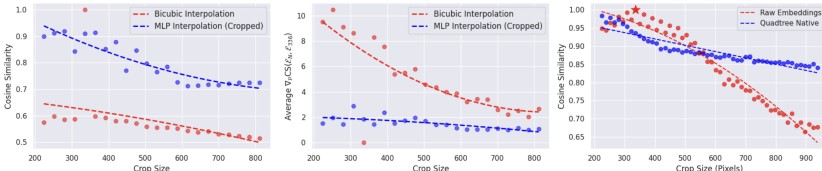

Figure 4: The first two panels compare our MLP interpolation with bicubic interpolation. We plot $C_{N\to336}^z$ in the first panel as a measure of mesoscopic bias and $\mathcal{B}_{\text{Interp}}$ in the middle panel as a measure of interpolation bias. The third panel shows a comparison between the [CLS] tokens of various image sizes with (blue) and without (red) QtP. All data is collected and averaged over the images from the $V^*$ benchmark.

## 4 EXPERIMENTAL RESULTS

Recall that the parameter $\alpha$ from equation 2 controls the amount of pruning done to the quadtree. For natural images, users can reuse the trained MLP and tune only $\alpha$; default values work reasonably well across similar domains. As Figure 6 illustrates, performance exhibits non-monotonic behavior with respect to $\alpha$ and image size. We recommend $\alpha > 0$ for fine-grained tasks and $\alpha \approx 0$ for general VQA. We perform sweeps in $\alpha$ and image size, over a suite of multi-modal benchmarks, in an attempt to understand the dynamics of our proposed methodology. For some benchmarks we additionally sweep native image resolution vs. cropped image resolutions. We report the best score from our sweeps in Table 1. We choose to look at the performance on $V^*$ (Wu & Xie, 2024), MM-Bench (Liu et al., 2024b), POPE (Li et al., 2023), CV-Bench (Tong et al., 2024), the visual portion of ScienceQA (Lu et al., 2022), MME (Fu et al., 2023), and the RealWorld-QA benchmark (xAI Team, 2024). We use VLM Eval (Duan et al., 2024) to do the evaluations on MM-Bench, POPE, ScienceQA, MME, and RealWorld-QA. We use a custom evaluation script to evaluate $V^*$ and CV-Bench. More details of our experimental setup are contained in Appendix D.1 and instructions to reproduce our experiments are contained in Appendix H.

**QLIP Reduces Measured Interpolation and Mesoscopic Bias:** In Figure 4 we plot a comparison between QLIP and the vanilla CLIP encoder using bicubic interpolation, which we found outperformed bilinear interpolation. We see that MLP training successfully reduces interpolation bias as measured by $\mathcal{B}_{\text{Interp}}$ (Figure 4, middle panel), and brings the cosine similarity between the [CLS] tokens together as predicted by our theoretical assumptions. Next, we observe that the quadtree selection mechanism mitigates the effects of mesoscopic bias by slowing the rate at which the cosine similarity of the CLIP [CLS] tokens diverge as a function of image size (Figure 4, rightmost panel).

Table 1: Performance comparison between LLaVA-QLIP and baseline LLaVA models. **Bold** highlights the better-performing variant of the same base model. Underlining denotes the best result across all models. An asterisk (*) indicates results obtained using cropped images. Performance increases and decreases are annotated in green and red, respectively.

| Model | $V^*$ | MM-Bench | POPE F1 | CV-Bench | Sci-QA | MME | RW-QA |
|---|---|---|---|---|---|---|---|
| | | | *VQA* | | | | |
| LLaVA-1.5-7b | 42.4 | **62.5** | 74.4 | 39.9 | **64.0** | 1207 | **49.0** |
| + QLIP | **53.4** | 59.7 | **79.6** | **40.2** | 63.5 | **1241** | 47.3 |
| | (+11.0) | (-2.8) | (+5.2) | (+0.3) | (-0.5) | (+34) | (-1.7) |
| LLaVA-1.5-13B | 45.0 | 67.4 | 82.4 | **61.6** | 67.8 | **1390** | 48.0 |
| + QLIP | **58.6** | **67.9**\* | **83.6** | 60.7\* | **67.9** | 1388\* | **49.4** |
| | (+13.6) | (+0.5) | (+1.2) | (-0.9) | (+0.1) | (-2) | (+1.4) |

**QLIP Significantly Improves the Detailed Visual Grounding on High-Resolution Images:** The $V^*$ benchmark (Wu & Xie, 2024) is a challenging, vision centric benchmark focused on fine-grained image understanding. This benchmark is particularly challenging for CLIP-based vision encoders because the questions are designed to be answered with access to the full-resolution image (see

Figure 1). Without access to all of the appropriate visual information the model is often reduced to guessing.

Figure 5 demonstrates that in the absence of the quadtree selection method, our MLP interpolation network already allows the model to effectively utilize all of the image tokens from the original image. We note that the 7B parameter is more robust to image sizes which were not seen during training than the 13B model. These results already indicate that there is a large performance gap that can be closed with minimal interventions, indicating that a significant portion of the poor performance on high-resolution image tasks can be explained simply by a lack of access to high-quality visual input signal (c.f. (Li et al.)). This result indicates that the CLIP encoder and LLaVA weights possess sufficient capacity to do VQA, but lack high-quality inputs.

Figure 6 shows the full sweep over image size and $\alpha$, plotted with cubic best-fit lines. The $x$-axis is measured in percentage of tokens seen compared to the baseline model, on a logarithmic scale. We see a clear trend where increasing $\alpha$ increases performance with maximal performance occurring for $\alpha > 0$. This indicates that the QtP mechanism is complementing the MLP interpolation to boost VQA performance, either by reducing the number of image tokens and sending stronger attention signal to the LLM (Levy et al., 2024; Veličković et al., 2024), by reducing noise by combining redundant image patches through merging, or both. Our ablations in Section 5 below suggest that the latter is more likely.

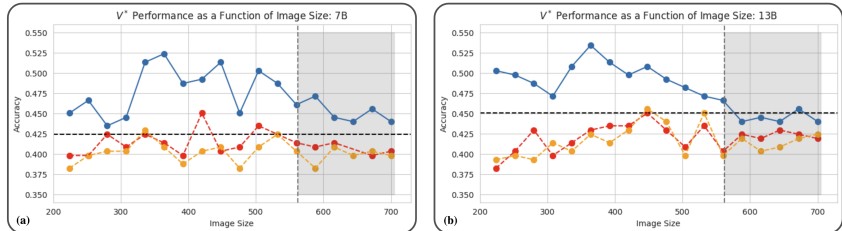

Figure 5: The performance on $V^*$ using re-scaled and cropped images with no quadtree selection mechanism and our MLP interpolation. The red line is with bicubic interpolation and the orange line is with bilinear interpolation. The black line represents performance of the base CLIP model with $336 \times 336$ cropping. The 7B model is plotted on the left, and the 13B model on the right. We see that neither bilinear nor bicubic interpolation is suitable for extending CLIP to larger resolutions.

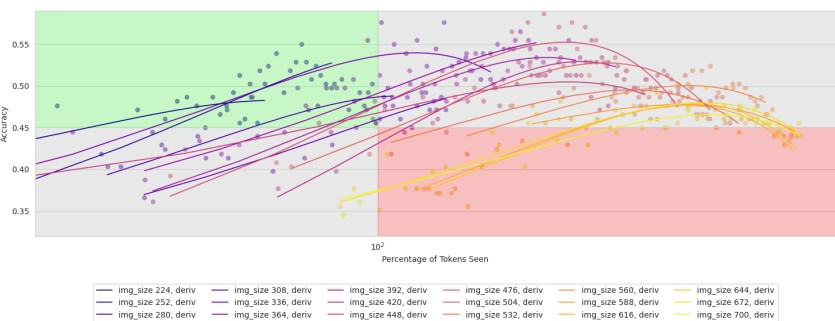

Figure 6: The compute vs. accuracy curves for our sweep of $V^*$ with the LLaVA-QLIP-13B model. The $x$-axis is on a logarithmic scale. The green-shaded region highlights experiments where our model **surpasses the baseline** with **fewer** visual tokens.

**Improved Token Efficiency:** Previous studies which reduce token counts have aimed at matching MLLM performance with fewer tokens (Cao et al., 2023; Chen et al., 2024; Hu et al., 2024; Li et al.; Tang et al., 2022). Our work is largely orthogonal to the aforementioned works, however we note in Figure 6 that we can achieve higher than baseline accuracy with *fewer* image tokens than the baseline model. This is shown in the figure by the top left region, shaded in green, which represents higher than baseline accuracy with lower than baseline numbers of tokens. This reveals that the quadtree selection method prunes tokens in such a way that higher-quality visual signal is provided to the LLM. The work (Li et al.) demonstrated that such improved performance with reduced tokens is

theoretically possible, but to our knowledge this work is the first time such a result has been achieved in practice.

**QLIP Matches or Improves Performance Across a Range of MLLM Benchmarks:** Because our method is trained to be both minimally invasive and does not require re-training of the MLLM, we can adjust the model parameters to fit the task at hand without re-training. Because our training program was oriented towards matching CLIP outputs on images which are the same size as CLIP was trained on, we can nearly achieve baseline performance for any benchmark by using $336 \times 336$ images with $\alpha = 0$. Any loss in performance beyond that can be attributed to the error in interpolating the CLIP embeddings with our MLP network. Notably we find little to no change in performance on MM-Bench, CV-Bench, Sci-QA, MME, or RealWorld QA.

QLIP is designed to perform well on images with fine-grained, high-resolution semantic detail. Because of this global or coarse cues useful for general VQA may be attenuated. For example, when tuned for fine-grained performance on $V^*$, the 7B model shows regressions on MM-Bench and RealWorld-QA (Table 1). The method is intentionally specialized for fine-grained tasks and reveals latent capacity in existing models rather than aiming to replace full retraining approaches.

**LLaVA 13B is Sensitive to Image Aspect Ratio:** On three of the seven benchmarks the 13B model attained its best performance when the input images were cropped to be square at the original image resolution of $336 \times 336$ with $\alpha < 0.1$. We found that performance quickly dropped off for these three benchmarks when we varied image size or increased $\alpha$. We suspect that the 13B parameter version of LLaVA is much more sensitive to deviations in the [CLS] token, and the drop-off in performance seems correlated with the change in cosine similarity of the [CLS] token plotted in Figure 4. We did not observe the same trend in the 7B model, nor did we observe this trend on $V^*$, where the content of the [CLS] token is not helpful for answering the questions.

Table 2: Comparison of LLaVA-QLIP with other models which improve fine-detail grounding. We report the numbers from the authors' papers. Note that $S^2$ requires pre-training and instruction tuning of the LLM Shi et al. (2024), and that SEAL requires fully re-placing the vision encoder before pre-training and instruction tuning Wu & Xie (2024).

| Model | $V^*$-Att | $V^*$-Rel | $V^*$ Overall | POPE F1 |
|---|---|---|---|---|
| *Fine-grained grounding* | | | | |
| QLIP-7B | 50.4 | 60.5 | 53.4 | 79.6 |
| $S^2$-7B Shi et al. (2024) | 51.3 | 61.8 | 55.5 | - |
| QLIP-13B | 53.9 | 65.8 | 58.6 | **83.6** |
| $S^2$-13B Shi et al. (2024) | 50.4 | 63.2 | 55.5 | - |
| SEAL (7B) Wu & Xie (2024) | **74.8** | **76.3** | **75.4** | 82.4 |

**Hallucination can be Mitigated by Reducing the Number of Image Tokens:** The POPE dataset was designed to measure the hallucination proclivity of MLLMs (Li et al., 2023). The proposed measurement of model performance for POPE is the F1 score. For both the 7B and 13B QLIP models we saw increased performance on POPE, with more significant gains for the 7B model. In fact, QLIP even outperforms SEAL (Wu & Xie, 2024) which is a heavily optimized version of LLaVA designed specifically to address fine-grained VQA (see Table 2). We found that peak POPE performance occurred with the smallest image size we tested (shortest edge is $224$ pixels), and an $\alpha = 0.7$, corresponding to slightly less than 50% of the baseline image tokens.

## 5 ABLATIONS

We ablate our design decisions along two axes. The first axis is along interpolation strategy, where we show that our MLP network vastly outperforms bilinear and bicubic interpolation. Next, we demonstrate that our performance improvements from the quadtree mechanism are predicated on selection strategy and not due solely to a reduced token counts. More detailed ablations are contained in Appendix G.

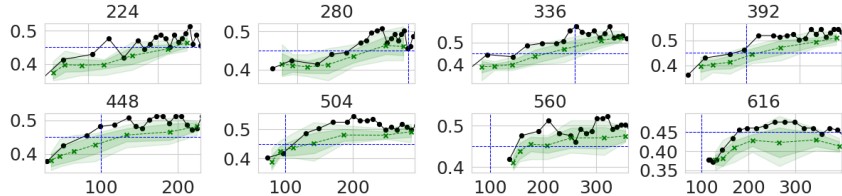

Figure 7: Ablation on $V^*$ with QLIP-13B. The black curves are QLIP, with derivative pruning, and the green curves are QLIP with random pruning. The green curves are plotted with min/max lightly shaded, and the first standard deviation more darkly shaded. Each of the evaluations with the random selection strategy was run 10 times to compute the average and standard deviation. The $x$-axis is the percentage of image tokens seen compared to baseline, and the $y$-axis is accuracy. Each pane is labeled with its image size and the vertical and horizontal blue dashed lines represent baseline number of image tokens and baseline accuracy respectively.

**MLP Interpolation is Essential for Generalizing to Arbitrary Image Sizes:** We experiment with using bicubic interpolation to scale evaluation with image size. We find that across all of our benchmarks bicubic and bilinear interpolation under-perform our MLP interpolation. This is clearly demonstrated for $V^*$ benchmark in Figure 5, where the bicubic and bilinear interpolation schemes under-perform even the baseline model performance on average.

**Performance Gains are not Solely a Result of a Reduced Number of Image Tokens:** We verify that the derivative selection strategy provides a meaningful information signal to the downstream LLM by comparing it to using a random selection strategy which prunes quadtree branches at some random rate. We compare the performance of these two selection strategies on the $V^*$ benchmark in Figure 7, where keeping precise semantic information about particular regions of the image is critically important. We find that on average there are large gaps in performance between random selection and derivative selection, indicating that our derivative selection strategy provides a more meaningful visual signal to the model.

## 6 RELATED WORK

**Improved Vision Encoders and MLLMs** The observation that grid patchification at a fixed image resolution is a poor inductive bias is not new. This has led to a litany of proposed replacements for CLIP (Radford et al., 2021) and ViT (Dosovitskiy et al., 2020). For example, the studies (Bolya et al., 2022; Darcet et al., 2023; Dehghani et al., 2023; Duggal et al., 2024; Fan et al., 2021; Haurum et al., 2023; Kong et al., 2022; Lee et al., 2022; Marin et al., 2023; Meng et al., 2022; Oquab et al., 2023; Yang et al., 2022; Zhang et al., 2022) propose modifications to the ViT architecture which provide better visual signal. These studies do not attempt to train an attendant LLM to create an MLLM. The studies (Bigverdi et al., 2024; Guo et al., 2024; Liu et al., 2024a; Lu et al., 2022; Luo et al., 2024; Shi et al., 2024; Thapa et al., 2024; Tong et al., 2024; Wang et al., 2024; Wu et al., 2024; Yang et al., 2022; Zhang et al., 2025) introduce new vision encoders specifically in the context of MLLM, but require pre-training and instruction tuning. The most closely related result work to ours is by Shi et al. (Shi et al., 2024) who show that LLaVA performance can be increased substantially by feeding the LLM visual tokens from different scales while keeping the CLIP encoder frozen. We go beyond all of these studies by obtaining improved performance using the *same* underlying MLLM backbone, with no pre-training, instruction-tuning, or supervised fine-tuning of the language model.

Newer vision encoders such as InternVL or Qwen-V employ integrated token reduction, multi-resolution training, or relative/rotary position encodings, which address interpolation and resolution handling in a different way than we have here. These architectural choices make direct integration of QLIP non-trivial, and we leave adaptation to such encoders as future work.

**Token Pruning and Merging** Many MLLM studies have been directed at reducing the number of visual input tokens, either by pruning tokens or merging them. Such reductions are well-motivated. (Levy et al., 2024; Veličković et al., 2024) show that in addition to being computationally expensive, feeding an LLM too many tokens can harm performance. Recent work has also demonstrated that MLLMs rely heavily on the `[CLS]` token during VQA (Zhang et al., 2024), which helps explain why previous authors have been able to remove up to 95% of the visual tokens and nearly maintain MLLM performance (Cao et al., 2023; Chen et al., 2024; Hu et al., 2024; Sun et al., 2025; Tang et al.,

2022), or prune tokens across video frames while maintaining performance (Choudhury et al., 2024). A key distinction is that QLIP operates *before* encoding, at the image patchification stage, whereas most pruning and merging methods operate *after* encoding and typically trade accuracy for efficiency. QLIP instead improves the signal quality during patchification rather than pruning encoded tokens. Additionally, all of these studies require an expensive pre-training and fine-tuning stage to align the LLM with the vision encoder. Furthermore, our work is orthogonal to the studies (Cao et al., 2023; Hu et al., 2024; Sun et al., 2025; Tang et al., 2022) since these models rely on training LLaVA family models while using the CLIP encoder, which can be replaced in their studies by QLIP.

## 7 CONCLUSION

We have proposed QLIP, a drop-in, adaptive, and content-aware replacement for the CLIP encoder. We defined mesoscopic bias and interpolation bias, argued that these biases are responsible for performance difficulties on fine-grained VQA, and shown that QLIP satisfactorily addresses these biases. We achieve +13.6% accuracy on the challenging $V^*$ benchmark without retraining the vision encoder or LLM. We are also able to exceed baseline performance on $V^*$ while using fewer image tokens. On other benchmarks, we show that we can nearly match or exceed baseline performance.

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
