# A  LIMITATIONS

**Scope and Applicability**    QLIP is not designed for dense prediction tasks such as segmentation, object detection, or visual grounding. It does not guarantee preservation of pixel-level spatial fidelity and is intended for semantic aggregation and token budgeting rather than dense spatial reasoning. We have not validated QLIP on non-natural image domains (e.g., medical imaging, satellite imagery). Integration with newer MLLMs that employ integrated token reduction, multi-resolution training, or relative position encodings (e.g., InternVL, Qwen-VL, SigLIP-style encoders) requires architectural changes and is left as future work.

**Technical Assumptions**    In Section 2 we assumed that the `[CLS]` token should be constant as a function of image size. This assumption, while stronger than the original implicit prior of CLIP, still lacks theoretical justification. It is easy to argue that a strong vision prior could be stated. The reason for this is that the CLIP encoder's understanding of an image may change as we scale image size, despite the theoretical alignment of the semantic content. For example, in the leftmost panel of Figure 2 we would not expect an image in which the elephant occupies 576 patches to have the same `[CLS]` embedding as the zoomed out version.

We did not fully sweep or optimize the MLP training due to compute limitations (c.f. Appendix C for a discussion of how we arrived at our hyper-parameters). Sweeping the MLP training hyper-parameters more fully would likely yield a better MLP model.

We trained the MLP on images that were smaller than or equal to 560 on their shortest edge. This was primarily an exercise in the tradeoff between batch-size and image-size. Training on larger images is preferable, but at the cost of smaller batch-sizes and significantly longer training times. We found that 560 was a happy medium for this trade-off. Future work could explore ways to train the MLP on very large images without actually loading the entirety of the image into memory. We believe such a methodology would be useful more broadly in the computer vision / multi-modal communities.

We also did not explore training the MLP on different datasets. While we believe that the content of the images is largely immaterial we suspect that the distribution of image sizes is quite important. We leave an investigation of this relationship to future work.

While we did explore multiple selection strategies (c.f. Appendix F), there is room for a more comprehensive and theoretically justified exploration of potential selection strategies. For example one could run a large-scale study correlating different selection methods with how well they find the "correct" tokens predicted by (Li et al.).

Finally, we could run our model through a more comprehensive suite of benchmarks to gain a more accurate sense of its performance.

# B  IMPLEMENTATION DETAILS

**Inference Overhead**    Quadtree construction overhead is small compared to the CLIP forward pass. Inference time is dominated by token count. The procedure can be optimized substantially in compiled implementations, and with KV caching, quadtree construction is amortized.

**Quadtree Construction**    To build a quadtree out of patches requires an image to be (a) square with (b) sidelengths consisting of $2^N$ patches for $N \in \mathbb{N}$. Obviously we are able to apply our methodology to images which are not of this size and we explain how we do so.

For concreteness, suppose we are given an image consisting of $M \times N$ patches. We can always center crop the images to the nearest patch size at a loss of at most 13 pixels. For this paper we first resize the smallest edge to our target size, then center-crop the image so that the longest side is also an integer number of patches.

Next, we find a grid of sub-images which maximally covers the original image, and where each of the sub-images in the grid is square with side lengths of $2^P$ for some $P$. For the remaining patches we leave them as is and pass their embeddings to the LLM. This process maximizes the number of patches in the image that can be subject to QtP. See Figure 8 for an example of this methodology applied to an image.

For natural images, users can reuse the trained MLP and tune only $\alpha$; tuning $\alpha$ is recommended when optimizing for a specific benchmark or resolution. For full implementation details see our code at https://github.com/KyroChi/qlip (and Appendix H).

## C  DETAILED MLP TRAINING

### C.1  HYPERPARAMETERS AND TRAINING SETUP

In our ad-hoc testing we quickly determined that for benchmark performance the MLP interpolation error was much more significant than the overall [CLS] embedding error. Therefore, our subsequent training experiments were targeted primarily at reducing MLP error.

We did not perform an extensive hyperparameter sweep over MLP architectures because the cost was prohibitive given our available compute. In what follows we describe our findings as we manually swept in individual directions to ablate our training hyperparameters.

- $L^2$ loss for [CLS] tokens is better than cosine similarity.
- $L^1$ loss for interpolation loss is better than $L^2$ loss. We found that the $L^2$ version of equation 4 was made smaller during training if we used the $L^1$ loss as the actual training target. We suspect that this is because the $L^2$ loss gets quite small (on the order of $10^{-5}$ to $10^{-7}$, and stops sending meaningful signal to the weights.
- We swept four orders of magnitude for $\gamma$, $\gamma = 10^3, 10^2, 10, 1, 0.75$. We found that $\gamma = 1$ produced the best results and had the most stable training dynamics.
- Training on larger images produces better results but is more computationally expensive. The larger images seemed to give better results but took too long to perform meaningful sweeps over. We opted for a small batch size of 14 since it accelerated training while continuing to produce satisfactory results. We arrived at this number by choosing the maximal image size we were willing to train on and then saturating the GPUs.
- Dynamics appear stable regardless of batch size. We found that even with a very small batch size of 1 or 2 the training remained stable.
- Depth 4 MLP is better than a depth 2 MLP. We found that increasing the MLP depth from 2 to 4 gave better results and faster convergence of the interpolation error. We did not try depths greater than 4.

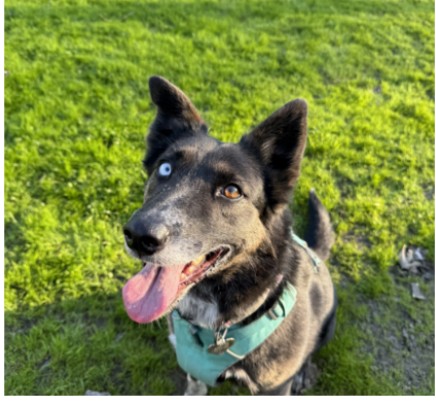 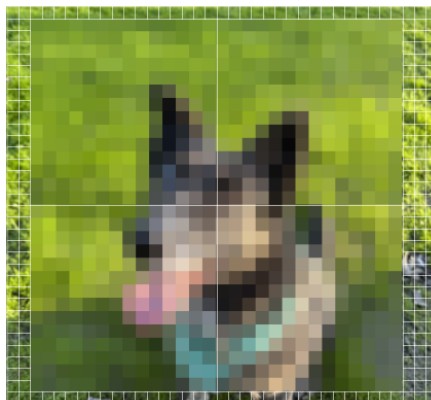

Figure 8: A quadtree applied to the image used in Figure 3, except with a different image size. The image in Figure 3 is $672 \times 896$, which can be decomposed into a $3 \times 4$ grid of $224 \times 224$ sub-images. Since $224 = 2^4 \times 14$ each of these 12 sub-images can have a QtP. The image in this figure is $476 \times 518$, which cannot be divided into QtP sub-images. The maximal grid of QtP-enabled sub-images is the $2 \times 2$ grid of $224 \times 224$ sub-images which are outlined in this Figure. Note the remaining patches are left as is around the border of the image.

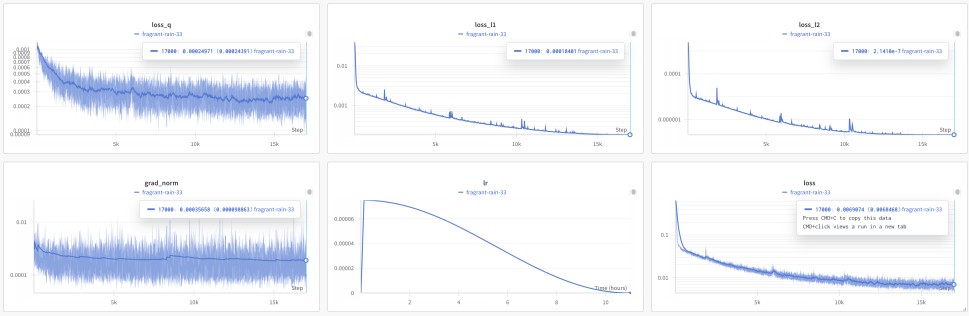

Figure 9: The training curves for our MLP training run. In the upper left is the $L_{\texttt{[CLS]}}$. The upper middle is the residual loss $\mathcal{R}$ from equation 4. The upper right is the $L^2$ analog of $\mathcal{R}$. Bottom left is the grad norm with respect to the positional encodings, given by $\mathcal{B}_{\text{Interp}}$ in equation 1 above. The bottom middle is our learning rate as a function of time. The bottom right is the training loss, which is the sum of the upper left and upper middle panels. We did not stop training when spikes occurred and we found that the loss spikes were transient. The upper row is plotted with a logarithmic $y$-axis, as are the bottom left and bottom right panels.

- Fourier features: We tried 16, 32, and 48 Fourier features and found that 48 yielded the best results.

- We used a cosine learning rate scheduler and did not experiment with adjusting the schedule. See our code for full details.

- Learning rate. We experimented with various learning rates and found that $7.5 \times 10^{-5}$ was a good learning rate. We experimented with higher learning rates and found that they made the training unstable.

- We use a hidden width of 1024 and did not experiment with other widths.

During training we use a learning rate of $7.5 \times 10^{-5}$ with the Adam optimizer using the default PyTorch configuration. Our MLP has four hidden layers and 48 Fourier features. We train for 100 epochs using a standard cosine learning rate scheduler. During training we use a quadtree with a $10\%$ random merging strategy. The reasoning for this is two-fold. First, introducing the random merging allows the model to see more patch locations during training, and thus effectively increases the image sizes that our model can handle. Second, it acts as a regularizer to prevent overfitting to the data-distribution of our training dataset. This is because with a deterministic sampler the positional encodings would align themselves to common QtP patterns. For example, if the objects in the training data were centered and the background had low semantic content, the model may overfit to such a situation and not be robust to situations in which the objects of interest are not centered in the image.

## C.2 FINAL TRAINING CURVES

The noise in the $L_{\texttt{[CLS]}}$ term and the grad norm terms is expected as a consequence of training on multiple resolutions simultaneously, as well as using the random selection strategy during training. We observed that the variance of these curves decreases if we restrict training to a narrower band of resolutions and / or remove the random selection from the training. The full training curve appears in Figure 9.

## C.3 DISCLOSURE OF ADDITIONAL COMPUTING RESOURCES

We report that training the MLP took 11 hours in Section 3.3. This time does not account for the hyperparameter sweeping that we did, nor does it account for the experimentation and development phase of our methodology. We did not keep track of the GPU hours that were used during the completion of this project. We had access to two 4x RTX 6000 machines, one 4x L40s machine, and one 8x RTX 6000 machine. We variously used compute on these three machines as it became available. Machines are shared between the members of our research group.

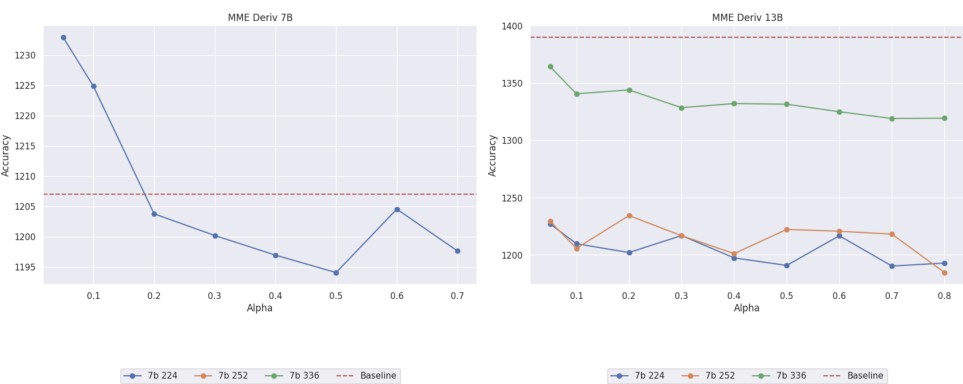

Figure 10: Sweeps on the MME benchmark (Fu et al., 2023). Baselines are indicated by the dashed red line.

# D    EXTENDED EXPERIMENTAL RESULTS

## D.1    MORE DETAILS ON EVALUATION STRATEGIES

We chose parameter sweep ranges through ad-hoc probing of both image size and $\alpha$ values. Once we found good enough endpoints we would sweep the values in-between, keeping the cost of evaluations in mind as we chose our sweep parameters. We would stop sweeping early if the results were trending in the wrong direction, since over regularization from QtP is expected to cause consistent declines in performance after a certain threshold.

## D.2    NATIVE IMAGE RESOLUTION VS. CROPPED

For models that did not perform well-enough using the native resolution we would switch to sweeping the cropped versions. In one case we report numbers from our model with no QtP and only the MLP interpolation (MME).

## D.3    MME BENCHMARK

We found that performance with native images was poor on MME so we swept cropped images. For the 7B model this lead to overperformance of the baseline, but for the 13B model we could not get overperformance of the baseline with $\alpha > 0$. Our top performing 13B model was with $336 \times 336$ image size, random selection, and $\alpha = 0$. This represents our model's closest approximation to the baseline model and any error is accounted for by the numerical error in the bicubic interpolation procedure. MME evaluations are expensive.

## D.4    MM-BENCH

For MM-Bench we found that the results were better with cropping than using the native image resolution. We swept several image sizes, but pruned the sweeps if the performance was proving poor. We swept image sizes of $224, 252, 336, 392$ and $448$. The results of our sweeps, including the specific $\alpha$ values chosen for each image size are shown in Figure 11.

## D.5    SCIQA

See Figure 12, Figure 12.

## D.6    POPE

See Figure 13. We abandoned sweeps which showed poor performance. POPE evaluations are expensive.

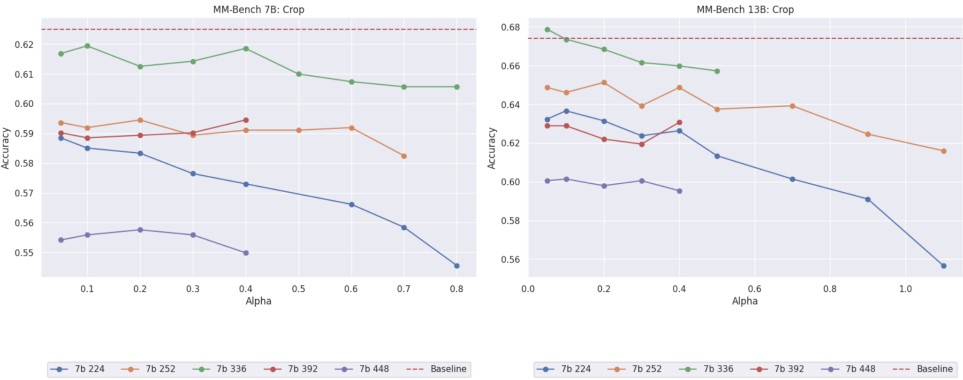

Figure 11: Sweeps on the MMBench benchmark (Liu et al., 2024b). Baselines are indicated by the dashed red line.

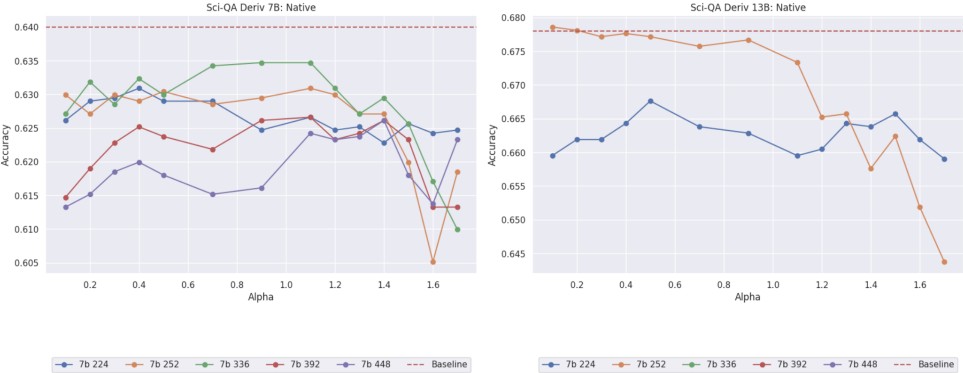

Figure 12: Sweeps on the ScienceQA benchmark (Lu et al., 2022). Baselines are indicated by the dashed red line.

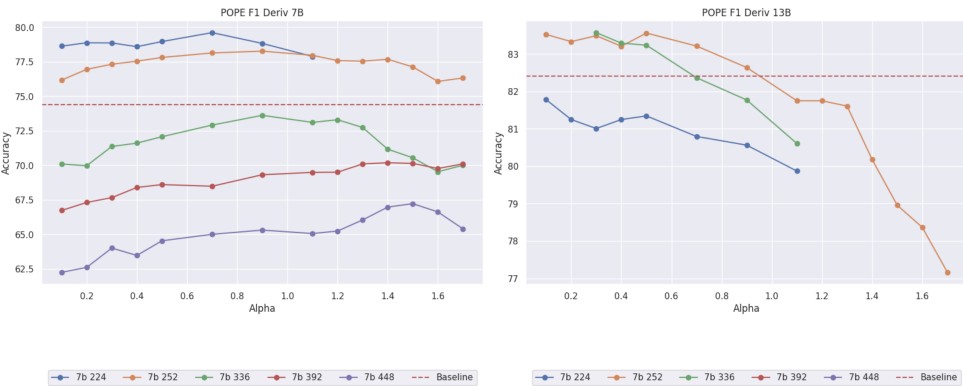

Figure 13: Sweeps on the POPE benchmark (Li et al., 2023). Baselines are indicated by the dashed red line.

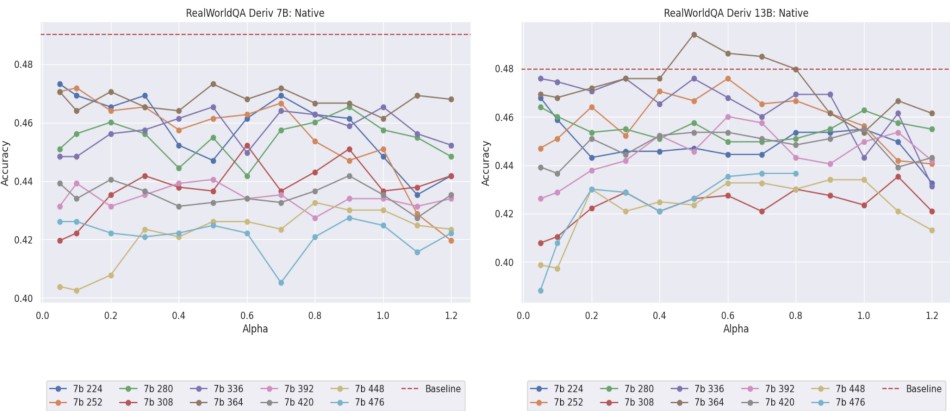

Figure 14: Sweeps on the RealWorldQA benchmark (xAI Team, 2024). Baselines are indicated by the dashed red line.

Figure 13

### D.7 REALWORLDQA

We are able to perform fairly comprehensive sweeps on the RealWorldQA benchmark (xAI Team, 2024) as the benchmark proves inexpensive to evaluate. The results of our sweeps on the 7B and 13B QLIP model are shown in Figure 14.

### D.8 CV-BENCH

CV-Bench (Tong et al., 2024) is expensive to run sweeps over. Because of this we searched a relatively small percentage of the search space. We found that performance using the 7B model was best for the larger image sizes (see Figure 15). We found that the QtP procedure typically led to decreasing performance on CV-Bench, and a preliminary sweep showed us that the 7B model performed best when using the larger image sizes.

For 13B the model performed poorly with the native image sizes and we swept crops instead. For this sweep we found smaller images were better, with peak performance occurring when the images matched the pre-training image size, $336 \times 336$ (see Figure 16).

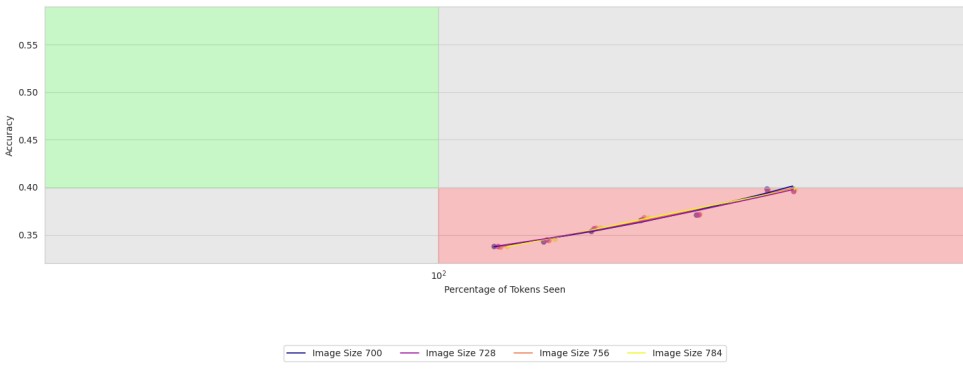

Figure 15: Compute vs. accuracy curves for our sweeps of CV-Bench, 7B, native resolution.

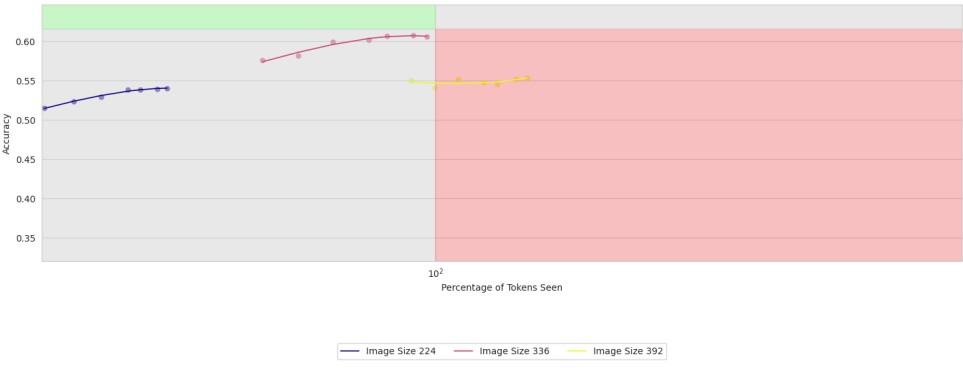

Figure 16: Compute vs. accuracy curves for our sweeps of CV-Bench, 13B, cropped resolution.

Table 3: Accuracy on $V^*$ for LLaVA-QLIP-7B using derivative selection method. Native resolution.

| Image Size | α=0.05 | α=0.10 | α=0.20 | α=0.30 | α=0.40 | α=0.50 | α=0.60 | α=0.70 | α=0.80 | α=0.90 | α=1.00 | α=1.10 | α=1.20 | α=1.30 | α=1.40 | α=1.50 | α=1.70 | α=1.90 | α=2.10 | α=2.50 | α=3.00 |
|---|---|---|---|---|---|---|---|---|---|---|---|---|---|---|---|---|---|---|---|---|---|
| 224 | 45.03% | 45.03% | 44.50% | 42.93% | 43.46% | 43.98% | 43.98% | 42.93% | 42.41% | 40.84% | 40.84% | 41.88% | 42.41% | 43.46% | 41.88% | 41.88% | 45.03% | 42.41% | 43.46% | 37.17% | 39.27% |
| 252 | 46.60% | 47.64% | 47.64% | 46.60% | 46.60% | 47.64% | 48.17% | 50.26% | 50.26% | 48.69% | 47.12% | 47.64% | 48.17% | 47.64% | 45.55% | 45.03% | 45.03% | 41.88% | 42.93% | 40.31% | 34.55% |
| 280 | 43.46% | 42.93% | 43.46% | 42.93% | 43.46% | 42.93% | 42.41% | 42.93% | 42.93% | 45.55% | 43.98% | 42.93% | 43.98% | 45.03% | 44.50% | 43.98% | 43.98% | 43.46% | 43.46% | 42.41% | 37.70% |
| 308 | 44.50% | 43.98% | 44.50% | 44.50% | 45.03% | 45.03% | 45.55% | 45.55% | 46.60% | 46.60% | 46.60% | 47.64% | 44.50% | 46.60% | 48.17% | 43.46% | 41.36% | 40.31% | 41.36% | 39.79% | |
| 336 | 51.31% | 51.31% | 51.31% | 51.31% | 49.74% | 47.12% | 47.12% | 47.12% | 47.12% | 48.17% | 49.21% | 48.69% | 48.17% | 47.64% | 46.60% | 47.12% | 50.26% | 44.50% | 45.03% | 40.84% | 36.65% |
| 364 | 52.36% | 52.36% | 52.36% | 52.88% | 53.40% | 52.88% | 52.36% | 51.83% | 53.40% | 51.31% | 50.79% | 49.21% | 47.12% | 51.83% | 50.79% | 50.79% | 47.64% | 46.60% | 43.46% | 41.88% | |
| 392 | 48.69% | 47.64% | 48.69% | 47.64% | 47.64% | 47.12% | 47.64% | 45.55% | 47.64% | 47.12% | 47.12% | 47.64% | 45.03% | 45.03% | 46.60% | 46.07% | 44.50% | 42.93% | 43.98% | 38.74% | |
| 420 | 49.21% | 49.21% | 49.74% | 50.79% | 51.31% | 49.21% | 50.26% | 46.60% | 47.64% | 47.12% | 49.21% | 50.26% | 49.21% | 47.64% | 46.60% | 45.55% | 49.21% | 45.03% | 42.41% | 40.84% | 37.70% |
| 448 | 51.31% | 50.79% | 51.31% | 52.36% | 52.88% | 51.31% | 51.31% | 51.31% | 50.26% | 49.21% | 48.17% | 49.21% | 47.64% | 48.17% | 49.74% | 46.60% | 43.46% | 45.55% | 46.60% | 42.93% | 39.79% |
| 476 | 45.03% | 46.60% | 46.60% | 45.55% | 45.55% | 44.50% | 47.12% | 44.50% | 46.60% | 46.60% | 48.69% | 48.69% | 45.03% | 45.55% | 42.93% | 44.50% | 45.55% | 41.88% | 41.88% | 38.22% | 36.65% |
| 504 | 50.26% | 50.26% | 51.31% | 52.88% | 51.83% | 51.31% | 52.36% | 52.36% | 51.83% | 49.74% | 51.31% | 48.69% | 48.69% | 51.31% | 49.74% | 51.31% | 50.26% | 49.21% | 46.60% | 43.98% | 40.84% |
| 532 | 48.69% | 49.74% | 50.79% | 49.21% | 48.17% | 48.69% | 50.26% | 50.79% | 48.17% | 47.64% | 48.69% | 49.74% | 49.74% | 48.17% | 45.55% | 45.03% | 46.07% | 49.21% | 46.60% | 42.93% | 38.22% |
| 560 | 46.07% | 45.03% | 46.60% | 45.55% | 44.50% | 42.93% | 43.46% | 45.03% | 44.50% | 42.41% | 42.93% | 42.93% | 43.46% | 45.55% | 46.60% | 47.64% | 46.07% | 47.12% | 44.50% | 45.03% | 41.88% |
| 588 | 47.12% | 46.60% | 46.60% | 46.60% | 48.17% | 48.17% | 49.21% | 48.69% | 47.12% | 46.60% | 46.07% | 47.12% | 48.69% | 46.60% | 50.26% | 50.26% | 46.07% | 46.60% | 48.69% | 42.93% | 47.64% |
| 616 | 44.50% | 42.93% | 43.98% | 45.55% | 46.07% | 44.50% | 46.07% | 46.07% | 47.64% | 47.64% | 47.64% | 46.60% | 46.60% | 46.07% | 47.64% | 46.60% | 46.07% | 44.50% | 42.93% | 47.64% | 37.70% |
| 644 | 43.98% | 43.46% | 44.50% | 45.55% | 46.07% | 45.55% | 46.07% | 47.12% | 46.07% | 45.55% | 49.21% | 47.64% | 45.03% | 45.03% | 48.17% | 46.60% | 42.41% | 35.60% | 37.17% | 37.17% | 37.70% |
| 672 | 45.55% | 45.03% | 45.55% | 45.55% | 47.12% | 47.64% | 47.64% | 49.74% | 47.12% | 45.03% | 47.64% | 47.12% | 42.93% | 46.07% | 45.55% | 42.93% | 40.31% | 37.70% | 35.08% | 37.70% | 35.60% |
| 700 | 43.98% | 45.03% | 44.50% | 45.55% | 45.55% | 47.12% | 46.60% | 47.64% | 47.12% | 45.03% | 45.03% | 44.50% | 42.93% | 43.46% | 40.31% | 44.50% | 39.79% | 41.88% | 36.13% | 34.55% | 34.55% |

Table 4: Accuracy on $V^*$ for LLaVA-QLIP-7B using random selection method. Native resolution.

| Image Size | α = 0.00 | α = 0.05 | α = 0.10 | α = 0.20 | α = 0.30 | α = 0.40 | α = 0.50 | α = 0.60 |
|---|---|---|---|---|---|---|---|---|
| 224 | **46.07%** | 43.46% | 40.31% | 40.84% | 40.84% | 40.31% | 34.03% | 35.60% |
| 252 | 45.03% | **47.12%** | 45.55% | 41.36% | 40.84% | 35.60% | 36.65% | 35.08% |
| 280 | 43.98% | 44.50% | 43.46% | **45.03%** | 39.79% | 40.31% | 42.93% | 40.84% |
| 308 | **45.55%** | 42.41% | 41.36% | 40.84% | 37.70% | 37.70% | 38.74% | 37.17% |
| 336 | **51.31%** | 46.60% | 45.55% | 43.98% | 41.36% | 35.60% | 38.74% | 37.17% |
| 364 | **52.88%** | 47.64% | 51.83% | 44.50% | 45.03% | 36.13% | 44.50% | 38.74% |
| 392 | 48.69% | **49.21%** | 46.60% | 42.93% | 42.41% | 41.88% | 43.46% | 33.51% |
| 420 | **49.74%** | 47.12% | 47.64% | 47.12% | 48.17% | 37.17% | 36.65% | 37.70% |
| 448 | **52.88%** | 50.79% | 45.55% | 43.46% | 40.84% | 37.17% | 39.27% | 38.22% |
| 476 | **46.07%** | 45.03% | 42.93% | 43.98% | 42.93% | 40.84% | 33.51% | 35.08% |
| 504 | **49.74%** | 47.64% | 48.17% | **49.74%** | 42.41% | 44.50% | 37.70% | 40.31% |
| 532 | 48.17% | **49.21%** | 47.12% | 48.17% | 42.93% | 42.41% | 39.27% | 41.88% |
| 560 | 46.07% | 45.55% | 43.98% | **48.69%** | 41.88% | 42.41% | 43.46% | 42.93% |
| 588 | 47.64% | 46.60% | **49.74%** | 45.55% | 49.21% | 46.07% | 43.98% | 46.07% |
| 616 | 44.50% | 44.50% | 41.88% | **45.55%** | 39.79% | 38.74% | 40.31% | 37.70% |
| 644 | 42.93% | 46.07% | 45.55% | **48.17%** | 35.60% | 42.41% | 39.79% | 37.70% |
| 672 | 45.03% | 45.03% | **47.64%** | 41.88% | 39.27% | 39.27% | 39.27% | 34.03% |
| 700 | 43.46% | **46.60%** | 46.07% | 39.79% | 41.88% | 40.31% | 39.79% | 35.08% |

## D.9 $V^*$-BENCH

For the $V^*$ benchmark (Wu & Xie, 2024) we sweep image size and $\alpha$ using our native image resolutions. We did not sweep $V^*$ using cropped images. We sweep image sizes between 224 and 700 in steps of 28. For the derivative selection strategy we sweep $\alpha \in (0.05, 0.1, 0.2, 0.3, 0.4, 0.5, 0.6, 0.7, 0.8, 0.9, 1.0, 1.1, 1.2, 1.3, 1.4, 1.5, 1.7, 1.9, 2.1, 2.5, 3.0)$. For the random selection strategy we sweep $\alpha \in (0.0, 0.05, 0.1, 0.2, 0.3, 0.4, 0.5, 0.6)$. $V^*$ evaluations are the least expensive of our chosen evaluations and therefore we have the most comprehensive sweeps on this benchmark.

We include the results of our sweeps in color coded tables below. The sweep for the 7B model with the derivative selection strategy can be found in Table 3. The sweep for the 7B model with the random selection strategy can be found in Table 4. The sweep for the 13B model with the derivative selection strategy can be found in Table 5. The sweep for the 13B model with the random selection strategy can be found in Table 6.

For the baseline model we sweep

## D.10 DISCLOSURE OF ADDITIONAL COMPUTING RESOURCES

We did not track the amount of time that our evaluation experiments took, although we plan to update this manuscript with this information once we have re-run the experiments. We had access to two 4x RTX 6000 machines, and one 8x RTX 6000 machine. We variously used compute on these three machines as it became available. Machines are shared between the members of our research group.

Table 5: Accuracy on $V^*$ for LLaVA-QLIP-13B using derivative selection method. Native resolution.

| Image Size | $\alpha = 0.05$ | $\alpha = 0.10$ | $\alpha = 0.20$ | $\alpha = 0.30$ | $\alpha = 0.40$ | $\alpha = 0.50$ | $\alpha = 0.60$ | $\alpha = 0.70$ | $\alpha = 0.80$ | $\alpha = 0.90$ | $\alpha = 1.00$ | $\alpha = 1.10$ | $\alpha = 1.20$ | $\alpha = 1.30$ | $\alpha = 1.40$ | $\alpha = 1.50$ | $\alpha = 1.70$ | $\alpha = 1.90$ | $\alpha = 2.10$ | $\alpha = 2.50$ | $\alpha = 3.00$ |
|---|---|---|---|---|---|---|---|---|---|---|---|---|---|---|---|---|---|---|---|---|---|
| 224 | 50.26% | 45.55% | 48.69% | 46.07% | 51.31% | 49.21% | 47.64% | 46.60% | 48.69% | 45.03% | 47.64% | 48.69% | 48.17% | 46.07% | 44.50% | 47.12% | 41.88% | 47.64% | 42.93% | 41.36% | 35.60% |
| 252 | 49.74% | 50.26% | 51.31% | 52.36% | 50.79% | 52.88% | 54.97% | 51.83% | 49.21% | 53.93% | 52.36% | 51.83% | 50.26% | 45.55% | 47.12% | 43.98% | 42.93% | 40.31% | 39.27% | 36.65% | |
| 280 | 48.69% | 49.74% | 48.69% | 46.07% | 45.55% | 50.26% | 49.21% | 47.64% | 49.21% | 48.17% | 47.12% | 50.79% | 50.26% | 49.74% | 47.64% | 47.12% | 44.50% | 43.98% | 41.36% | 42.41% | 40.31% |
| 308 | 47.12% | 48.69% | 47.12% | 48.17% | 47.12% | 49.21% | 49.21% | 47.64% | 45.03% | 48.69% | 46.07% | 47.64% | 45.55% | 47.12% | 45.03% | 42.93% | 41.36% | 39.79% | 41.36% | 36.65% | |
| 336 | 50.79% | 50.79% | 51.83% | 53.40% | 52.88% | 52.36% | 57.59% | 54.45% | 52.36% | 55.50% | 53.93% | 54.97% | 57.59% | 55.50% | 50.79% | 49.74% | 49.74% | 48.69% | 43.46% | 44.50% | 37.70% |
| 364 | 53.40% | 55.50% | 56.54% | 55.50% | 54.45% | 54.45% | 54.97% | 53.93% | 53.93% | 53.93% | 53.93% | 50.26% | 48.17% | 50.26% | 50.26% | 51.31% | 49.74% | 47.64% | 41.88% | 38.74% | |
| 392 | 51.31% | 55.50% | 53.40% | 54.45% | 54.45% | 52.88% | 51.31% | 54.45% | 52.36% | 54.45% | 50.79% | 50.26% | 52.36% | 51.83% | 51.31% | 51.83% | 51.83% | 46.07% | 44.50% | 42.93% | 36.13% |
| 420 | 49.74% | 52.88% | 53.40% | 52.88% | 55.50% | 52.88% | 53.40% | 52.88% | 51.31% | 52.88% | 51.83% | 51.31% | 52.88% | 52.88% | 51.83% | 49.74% | 48.17% | 46.07% | 44.50% | 39.79% | 37.70% |
| 448 | 50.79% | 49.74% | 50.79% | 51.31% | 47.64% | 47.12% | 48.69% | 51.31% | 51.31% | 49.21% | 51.31% | 51.31% | 50.26% | 47.64% | 48.17% | 50.79% | 48.69% | 48.17% | 45.55% | 42.41% | 37.70% |
| 476 | 49.21% | 51.31% | 51.31% | 49.74% | 49.21% | 49.74% | 51.83% | 52.36% | 52.36% | 55.50% | 57.07% | 57.59% | 57.07% | 58.64% | 57.59% | 56.54% | 49.74% | 45.55% | 47.12% | 40.84% | 39.27% |
| 504 | 48.17% | 48.17% | 49.74% | 51.31% | 49.74% | 50.79% | 51.83% | 49.74% | 50.79% | 49.74% | 51.83% | 53.40% | 52.88% | 52.88% | 54.45% | 52.36% | 52.36% | 50.26% | 48.69% | 41.88% | 40.31% |
| 532 | 47.12% | 45.55% | 48.69% | 48.69% | 46.07% | 46.60% | 46.60% | 48.69% | 48.17% | 49.21% | 48.69% | 48.69% | 49.21% | 49.21% | 46.07% | 47.64% | 48.17% | 51.31% | 47.64% | 44.50% | 41.88% |
| 560 | 46.60% | 47.64% | 48.69% | 49.74% | 50.26% | 50.26% | 49.74% | 49.21% | 52.36% | 51.83% | 51.83% | 48.69% | 48.17% | 49.21% | 46.07% | 47.64% | 48.17% | 51.31% | 48.69% | 47.64% | 41.88% |
| 588 | 43.98% | 43.98% | 44.50% | 47.12% | 46.60% | 48.69% | 48.17% | 46.07% | 46.07% | 49.74% | 45.55% | 45.03% | 46.60% | 46.07% | 46.60% | 47.64% | 43.98% | 47.12% | 42.93% | 41.36% | 41.88% |
| 616 | 40.84% | 42.41% | 43.46% | 40.84% | 46.07% | 45.03% | 45.55% | 47.12% | 46.60% | 49.74% | 47.64% | 46.60% | 47.12% | 47.64% | 42.41% | 43.98% | 37.70% | 40.31% | 37.17% | 37.70% | 38.22% |
| 644 | 42.41% | 42.41% | 43.98% | 43.98% | 42.93% | 45.55% | 46.60% | 44.50% | 46.07% | 46.60% | 45.03% | 45.55% | 43.98% | 45.55% | 44.50% | 43.46% | 44.50% | 42.41% | 39.27% | 36.65% | 37.17% |
| 672 | 41.88% | 45.03% | 43.46% | 42.93% | 43.98% | 47.12% | 47.12% | 46.60% | 49.21% | 48.17% | 47.64% | 46.60% | 43.98% | 47.12% | 47.64% | 42.93% | 43.98% | 40.84% | 38.74% | 38.22% | 35.08% |
| 700 | 43.46% | 42.41% | 46.07% | 45.03% | 43.98% | 47.12% | 43.98% | 46.07% | 45.55% | 46.07% | 47.12% | 44.50% | 46.07% | 44.50% | 42.41% | 44.50% | 45.03% | 40.31% | 39.79% | 36.65% | 35.60% |

Table 6: Accuracy on $V^*$ for LLaVA-QLIP-13B using random selection method. Native resolution.

| Image Size | $\alpha = 0.00$ | $\alpha = 0.05$ | $\alpha = 0.10$ | $\alpha = 0.20$ | $\alpha = 0.30$ | $\alpha = 0.40$ | $\alpha = 0.50$ | $\alpha = 0.60$ |
|---|---|---|---|---|---|---|---|---|
| 224 | 49.21% | 45.55% | **51.83**% | 43.46% | 45.55% | 39.27% | 38.22% | 36.65% |
| 252 | **49.74**% | 42.41% | 46.60% | 44.50% | 48.17% | 38.22% | 36.13% | 39.27% |
| 280 | **48.17**% | 45.55% | 42.41% | 42.93% | 40.84% | 38.74% | 37.70% | 39.27% |
| 308 | **49.21**% | 44.50% | 45.55% | 43.46% | 42.41% | 41.36% | 37.17% | 34.03% |
| 336 | 51.31% | **55.50%** | 48.17% | 45.55% | 41.36% | 40.31% | 39.79% | 40.31% |
| 364 | **53.93**% | 51.83% | 50.79% | 45.55% | 41.88% | 38.74% | 39.79% | 38.74% |
| 392 | **53.40**% | **53.40**% | 48.69% | 45.03% | 44.50% | 36.65% | 39.79% | 40.31% |
| 420 | 50.26% | 48.69% | **52.88**% | 47.12% | 48.69% | 42.41% | 40.84% | 34.55% |
| 448 | 50.26% | **51.83**% | 45.03% | 43.98% | 46.07% | 41.36% | 40.84% | 37.70% |
| 476 | **49.74**% | 47.64% | 46.60% | 48.17% | 47.12% | 47.12% | 39.27% | 40.84% |
| 504 | 49.21% | 48.17% | **49.74**% | 45.55% | 45.03% | 36.65% | 42.41% | 36.65% |
| 532 | 48.17% | 48.69% | 48.17% | **50.26**% | 44.50% | 42.93% | 40.31% | 40.31% |
| 560 | **46.60**% | 45.55% | 46.07% | 46.07% | 46.07% | 42.41% | 43.46% | 44.50% |
| 588 | 45.03% | 43.46% | 45.55% | **48.69**% | 45.55% | 43.98% | 44.50% | 45.03% |
| 616 | 38.22% | 42.93% | **45.55**% | 44.50% | 43.98% | 41.36% | 40.84% | 38.22% |
| 644 | 40.84% | 40.84% | **47.64**% | 41.88% | 40.31% | 40.31% | 39.79% | 38.22% |
| 672 | 39.79% | 42.41% | **42.93**% | 41.36% | 39.27% | **42.93**% | 42.41% | 37.17% |
| 700 | 41.36% | 40.31% | 42.41% | 42.93% | **43.98**% | 39.27% | 40.31% | 35.08% |

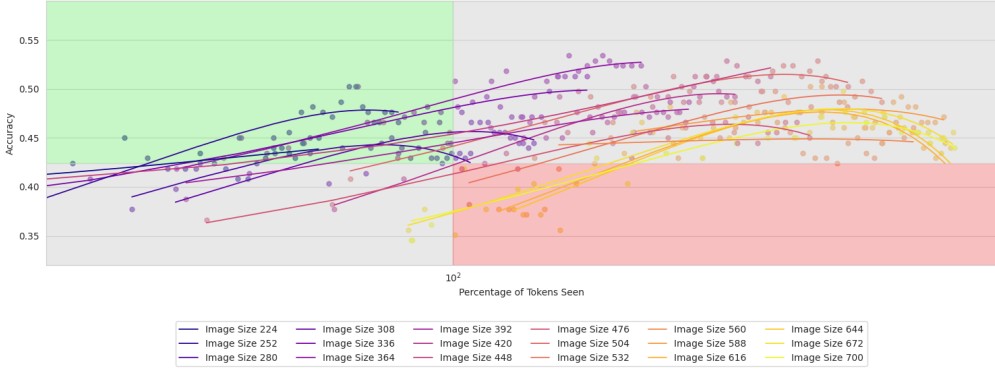

Figure 17: The compute vs. accuracy curves for. our sweep of $V^*$ with the LLaVA-QLIP-7B model. The $x$-axis is on a logarithmic scale. The green-shaded region highlights experiments where our model **surpasses the baseline** with **fewer** visual tokens.

# E  WHY WE DID NOT TO STUDY QWEN

The QWEN family of models (Yang et al., 2024) includes a vision transformer which is trained from scratch to handle arbitrary resolutions, so the MLP interpolation scheme is not necessary. QWEN implements 2D RoPE (Heo et al., 2024; Su et al., 2024) which can be interpolated natively by design. We attempted to apply the quadtree selection mechanism to the QWEN vision transformer but we were stopped by particularities in the QWEN model's token merging strategy. In particular they merge adjacent patches before feeding the patches into the vision encoder, which violates the inductive assumptions of the quadtree selection mechanism.

# F  QUADTREE SELECTION STRATEGY

We found that the directional derivative presented above outperformed more traditional measures like $\max_{x,y}(|\partial_x I| + |\partial_y I|)$. We do not have an explanation as to why this occurs. It is possible that the averaging strategy is better correlated with patches of interest than looking at the absolute magnitude. We additionally tested variance based methods during the exploratory phase of this project and found that they underperformed our derivative selection strategy.

## F.1  RANDOM PRUNING

Our quadtree implementation works from the root down and decides whether or not to split or not by looking at the split condition. Because we apply quadtree to sub-images of size $2^N \times 2^N$, each sub-image can also be quadtree patchified. To implement random pruning we decide to split a given node with probability $p$, sampled from a uniform distribution.

# G  DETAILED ABLATIONS

We plot a more comprehensive ablation sweep over $V^*$ than was provided in Figure 7 above. Figure 18 is the ablation for the 7B model on all of the image sizes and values of $\alpha$ that we tested. Figure 19 is the ablation for the 13B model on all of the image sizes and values of $\alpha$ that we tested.

# H  REPRODUCIBILITY

Our code is available at `https://github.com/KyroChi/qlip`. We release all of our code, including training code for the MLP and evaluation code for the trained model. Additionally, we will release our model weights which were used for this paper and also the results that we obtained for this paper.

In our codebase there will be a single command which will

1. Run the training script to train an MLP network using the hyperparameters described in this paper.
2. Run the entire sweep over all of the evaluation benchmarks to reproduce the model results.

We alternatively include scripts to run reduced evaluations, which are far less computationally expensive than a full parameter sweep. In particular we run the evaluation for the best model that we found for each dataset (Table 1)

# I  LLM STATEMENT

We did not use LLMs in a significant way to aid our research during the completion of this work. Our LLM usage did not extend beyond using code assistants like copilot and for polishing the writing in our manuscript.

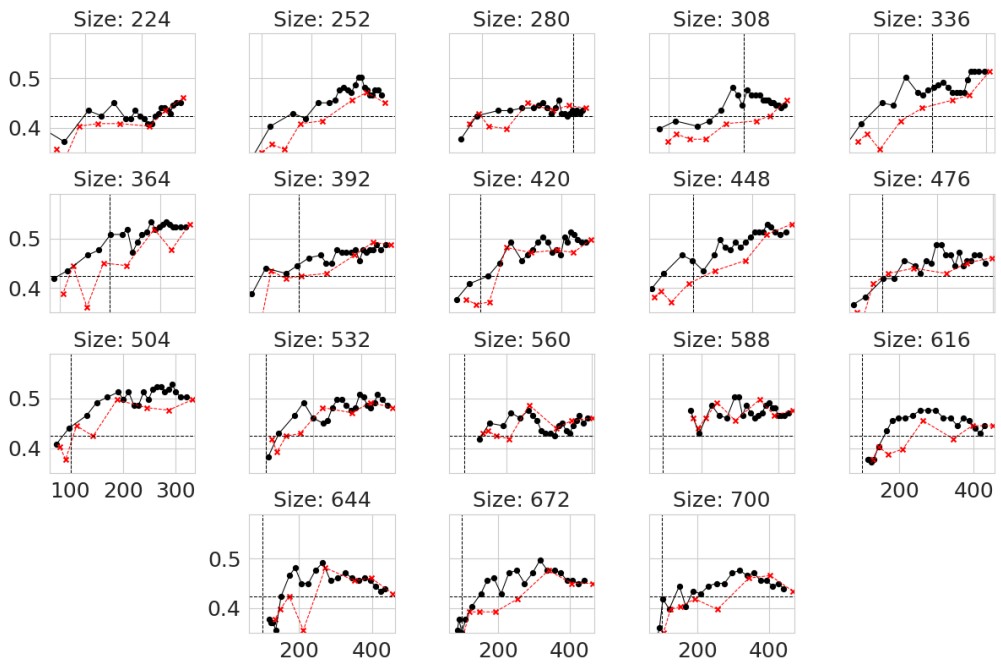

Figure 18: Ablation across a diverse range of image sizes of the QLIP-7B model on the $V^*$ dataset. The Black line is the QLIP performance with the derivative selection strategy and the red line is a random selection strategy. Each random selection trial was only run once.

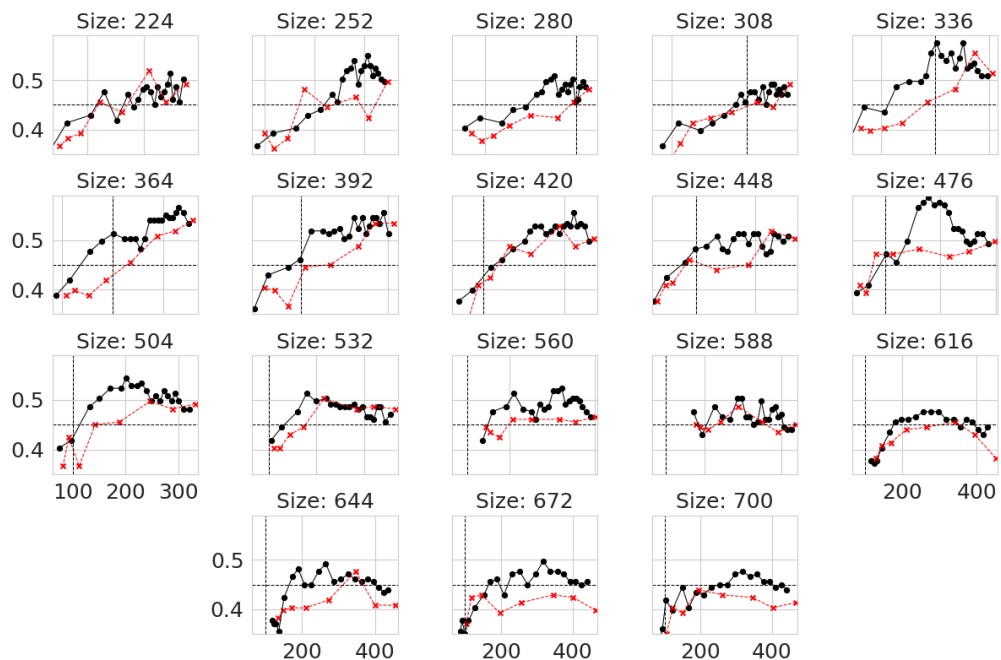

Figure 19: Ablation across a diverse range of image sizes of the QLIP-13B model on the $V^*$ dataset. The Black line is the QLIP performance with the derivative selection strategy and the red line is a random selection strategy. Each random selection trial was only run once.