# OpenReview forum: "QLIP: A Dynamic Quadtree Vision Prior Enhances MLLM Performance Without Retraining"
_ICLR.cc/2026/Conference — ICLR 2026 Poster_

### Official Review · Reviewer_5fie · 2025-10-30

**Soundness:** 3
**Presentation:** 3
**Contribution:** 2
**Rating:** 6
**Confidence:** 3

**Summary:**

This work proposes Q-LIP, a plug-and-play CLIP replacement that integrates into MLLMs with minimal code (no full retraining) and uses content-aware quadtree patchification to boost visual understanding.​ Experiments show QLIP improves LLaVA-1.5’s performance in several benchmarks, without full MLLM retraining/fine-tuning.

**Strengths:**

1. QLIP fixes CLIP’s mesoscopic/interpolation biases with quadtree patchification and a small MLP, handling arbitrary resolutions without encoder weight changes.
2. As a drop-in CLIP replacement, it boosts LLaVA’s  benchmark accuracy by 13.6% without MLLM retraining/fine-tuning.
3. QLIP improves token efficiency (higher accuracy with fewer tokens) and reduces hallucination, performing well on several benchmarks.

**Weaknesses:**

1. The application of the proposed method is limited. Currently, many large models natively support input image at arbitrary resolutions.
2. The generalization of the method is not validated. This study only conducts experiments on LLaVa-1.5 (7B and 13B), and experiments on additional models should be added.
3. It is advisable to add comparisons between the proposed quadtree patchification and some token merging/pruning methods.

**Questions:**

Refer to weaknesses.

---

> ### Author Response · Authors · 2025-12-04
> **Discussion / Rebuttal - Review 4**
>
> First, we want to thank the reviewer for taking the time to read our work and provide feedback. This year was plagued by bad reviews but we feel that this review was of high quality and made in good faith.
>
> The reviewer highlights several strengths of our work, including our approach to mitigating CLIP biases and that we do not have to re-train the underlying LLM.
>
> We have the following comments with regard to the weaknesses that they suggest:
> 1. While true that an increasing number of models supports arbitrary resolution images, this is not the only benefit of our method. As we show in Figures 5 and 6, our method is also able to obtain increases in performance by adaptively downsampling regions of the image. We show that reducing the visual token inputs to the model is beneficial beyond simply being able to handle arbitrary resolutions, and we hypothesize that the notion of mesoscopic bias will apply to other vision encoders. Thus, our work is well-situated to be applied in the context of other vision encoders.
>
> 2. We agree that it would be nice to extend this line of work to other models and other problems in future work. Unfortunately many modern MLLMs like QWEN or InternVL use complicated vision encoding solutions which add more implicit biases to the vision encoder. Thus it is not a simple matter of repurposing our approach. Rather, we think that the ideas we present here should be considered as well-reasoned approach to building new vision encoders for MLLMs. We also think that our work can inspire researchers to consider more deeply the implications of visual biases on VQA performance.
>
> 3. We caution that comparisons with token pruning and merging methods are challenging. Our method is neither a token pruning method nor a token merging method. Most existing pruning and merging methods act on vision tokens after they have been encoded, but our method acts on the images prior to encoding. With that being said, we are aware of no pruning or merging methods which report **increased** performance on any benchmark, making our method the only one that can use fewer tokens while also improving performance.
>
> In summary, the ability of our method to use fewer tokens and achieve higher performance is one of the ways in which our method differentiates itself from existing pruning and merging methods.

---

### Official Review · Reviewer_a7Xx · 2025-10-31

**Soundness:** 3
**Presentation:** 3
**Contribution:** 3
**Rating:** 6
**Confidence:** 4

**Summary:**

This paper introduces QLIP, a novel, lightweight, and drop-in modification for the CLIP vision encoder designed to enhance the performance of Multimodal Large Language Models (MLLMs), such as LLaVA, without requiring expensive retraining of the entire MLLM backbone. The authors attribute limitations of the standard CLIP encoder—specifically its fixed-resolution nature and poor handling of fine-grained details—to two biases: Mesoscopic Bias and Interpolation Bias. QLIP addresses these issues via two components: Quadtree Patchification (QtP) and Coordinate-Based MLP. Experiments show that QLIP achieves substantial performance improvements, notably a +13.6% accuracy boost on the challenging $V^{*}$ benchmark for fine-grained VQA, and successfully mitigates model hallucination.

**Strengths:**

1. Practical & Cost-Effective Drop-in Solution: QLIP’s design as a "drop-in replacement" is highly impactful. By significantly enhancing the visual signal without necessitating the re-training or fine-tuning of the entire MLLM pipeline, it offers a practical, low-cost path to upgrading existing MLLMs.

2. Clear Theoretical Motivation: The paper clearly and quantitatively identifies two specific, fundamental biases in the CLIP vision encoder (mesoscopic and interpolation bias). The proposed solutions are elegantly tailored to directly address these theoretical deficiencies.

3. Content-Aware Patchification: The use of Quadtree Patchification (QtP) with a simple gradient-based criterion provides an effective, training-free mechanism for adaptively merging semantically similar regions. This mechanism also has the beneficial side effect of reducing input token count, which is shown to decrease hallucination.

**Weaknesses:**

1. The base model is out of data. LLaVA suffers limited performance. I suggest the authors to conduct evaluation on more SOTA models, such as InternVL-3.5 or QwenVL2.5 to truly demonstrate the effectiveness.

2. I wonder the performance on visual grounding benchmarks, such as refcoco, since it also requires fine-grained region information.

**Questions:**

The QtP process involves a dynamic computation (quadtree traversal and gradient calculation). Can the authors quantify the computational overhead of the QtP stage in terms of wall-clock time? How does this overhead scale with very high input resolutions? Is the total prefill time (QtP + CLIP encoding) still significantly faster than a simple, high-resolution baseline that uses linear interpolation?

---

> ### Author Response · Authors · 2025-12-04
> **Discussion / Rebuttal - Review 3**
>
> First, we would like to thank the reviewer for taking the time to read our work and provide feedback. In a review cycle shadowed by controversy we appreciate the effort that went into writing this review.
>
> The reviewer highlights the practical aspects of our methodology, especially the fact that we do not need to re-train the underlying LLM. They also point out the clear motivation for our work, as well as highlighting the advantages of an adaptive method like ours.
>
> ### Weaknesses
> The reviewer suggests that we extend our results to InternVL-3.5 or QwenVL2.5. We agree that in the rapidly changing landscape of MLLMs it would be interesting to study these more recent models. We did look into adapting SoTA models after completing our LLaVA implementation, however both InternVL and QWEN use built-in token reduction methods in their vision encoders which makes implementing a quadtree patchification challenging. QWEN, for example, embeds adjacent image patches together, which breaks the inductive biases of the quadtree approach. Both InternVL and QWEN introduce additional visual biases which are not so easily identifiable or addressable as the visual biases in CLIP.
>
> We believe that our work also serves as a demonstration of the effect that implicit biases can have on a vision encoder's performance, and that this philosophy may motivate future improvements in state of the art models.
>
> ### Questions
> The reviewer rightly asks about the wall clock time of our method. The reason that we did not report these metrics is that we wanted our paper to focus on the theoretical benefits of using a quadtree patchification procedure and not focus on the wall-time optimization. The first author has worked on closed source quadtree implementations which were over 5000x faster than the Python implementation we used in this paper. We are confident that with sufficient engineering efforts the quadtree procedure can be made negligible compared to the forward pass through the neural network. Also note that with KV-caching, the quadtreeification only needs to be done once. In practice we found that inference speed was dominated by token count rather than the quadtree procedure, even for our Python implementation.

---

### Official Review · Reviewer_cr9P · 2025-10-31

**Soundness:** 3
**Presentation:** 3
**Contribution:** 2
**Rating:** 4
**Confidence:** 3

**Summary:**

CLIP is employed as the vision encoder for many MLLMs, yet its absolute positional encoding and fixed-resolution pre-training severely degrade performance when images are supplied at non-native resolutions.
The paper introduces QLIP, a drop-in replacement that can be inserted into any CLIP-based MLLM without architectural changes.  Instead of uniform patchification, QLIP first performs a content-aware partition that keeps salient regions at higher resolution and fuses the resulting variable-length token sequence into the original embedding dimension.  Extensive experiments show up to 13.6 % absolute improvement on a battery of cross-resolution benchmarks.

**Strengths:**

Simple, training-free at inference time, and demonstrably effective.

**Weaknesses:**

**Rapidly moving baseline.**  The community is already shifting from CLIP to newer vision backbones (e.g. InternVL, SigLIP-2) that use RoPE or 2-D absolute + relative encoders and are pre-trained with native multi-resolution recipes.  It is unclear whether QLIP retains any advantage when the underlying encoder itself is resolution-robust.  A head-to-head comparison with such models is missing.

**Limited to CLS-level bias.**  QLIP is optimised to deliver a single, high-quality CLS token; it does not guarantee per-token fidelity.  For dense-prediction tasks (segmentation, object detection) that require spatially accurate patch-wise features, the content-aware patchification may discard positional details and harm downstream accuracy.

**Questions:**

No

---

> ### Author Response · Authors · 2025-12-04
> **Discussion / Rebuttal - Review 2**
>
> First off, we thank the reviewer for taking the time to read our paper and provide meaningful feedback. In a year plagued with AI reviews we appreciate what appears to be a good faith review of our work.
>
> The reviewer highlights the demonstrable effectiveness of our approach, and we have the following comments on their proposed weaknesses of our work:
>
> 1. The reviewer points out that in the rapidly evolving field of AI, techniques which were in broad use at the start of 2025 may no longer be in broad use at the end of 2025. We agree that this represents a difficulty in scholarship; by the time research has been completed the base model may look like an anachronism.
>
>     Despite this, we believe that the combination of LLaVA and CLIP continues to provide fertile ground for MLLM research, and that our contributions provide meaningful insight into a bias common in **any** MLLM employing grid-based patchification.
>
>     With respect to RoPE: RoPE addresses the interpolation bias natively, which represents an improvement upon CLIP. One can generalize RoPE to apply to any patchification scheme, and indeed we have performed this calculation. However, we found that RoPE enabled vision encoders like the one used in QWEN have other implicit biases preventing the implementation of quadtree patchification, and training CLIP with RoPE requires completely re-training both CLIP and the underlying LLM.
>
>     We believe our work remains valuable even in light of shifting preferences on vision encoders. CLIP remains widely in use in industry, even if research is moving away from it. And it remains a solid foundation on which to explore patch-based vision encoders.
>
>     In summary, while it is true that the field is moving quickly, we believe that our work provides a perspective and potential solution for some of the biases present in vision encoders, including more recently released vision encoders.
>
> 2. The reviewer states that “QLIP is optimised to deliver a single, high-quality CLS token; it does not guarantee per-token fidelity” and that “For dense-prediction tasks (segmentation, object detection) that require spatially accurate patch-wise features, the content-aware patchification may discard positional details and harm downstream accuracy”. We do not agree with the first statement, and partially disagree with the second.
>
>     For the first statement, we note that the per-token fidelity is either the same as, or better than CLIP, which always resizes the image down first. Thus, QLIP can be set to guarantee that the highest level semantic patches match **pixel-for-pixel** the patches generated by CLIP. Our method retains visual semantic information better than standard CLIP, it is not purely optimized to deliver higher-quality CLS tokens, although it does also give higher-quality CLS tokens.
>
>     Regarding the discarding of positional details: The strength of our method is also a weakness in certain contexts, although we think that the reviewer’s statement can be bolstered by some nuance.
>
>     Our method is not intended to be good at segmentation tasks per-se. One salient aspect of our thesis is that a large amount of visual information is irrelevant for visual tasks. Our impression is that segmentation tasks do not suffer from the same issues as VQA with respect to the image inputs: segmentation can be done at the pixel level with high performance, which is manifestly untrue of VQA.
>
>     We specifically proposed a method to address CLIP biases, and more generally patch-based vision encoder biases, so our method has potential applications in patch-based semantic segmentation. In this case we hypothesize that our method will correlate with adjacent patches being assigned to the same semantic grouping.
>
> In summary, we believe that our work remains relevant despite the rate at which the field is progressing. We think that the characterization of our method as solely optimizing for the `[CLS]` token is not accurate. Finally, the reviewer is concerned about generalization to image segmentation. We agree with the reviewer that it is not obvious that our method is applicable to object segmentation but we make no claims that our method will generalize to this domain, which is far outside the scope of the present work.

---

### Official Review · Reviewer_y52t · 2025-11-01

**Soundness:** 3
**Presentation:** 3
**Contribution:** 3
**Rating:** 6
**Confidence:** 3

**Summary:**

This paper targets the limitations of the standard CLIP vision encoder used in Multimodal Large Language Models (MLLMs), specifically its fixed-resolution training and uniform patch grid1. The authors diagnose two key issues: "mesoscopic bias" (a preference for features at a specific training scale ) and "interpolation bias" (an inability to generalize positional encodings to new resolutions ). They propose QLIP, a "drop-in" replacement that combines two components: a content-aware "quadtree patchification" (QtP) mechanism to dynamically merge low-information patches based on image gradients , and a small, trained MLP to interpolate positional encodings for arbitrary coordinates. The central claim is that QLIP can be integrated into existing MLLMs like LLaVA without retraining the full model , leading to significant performance gains, most notably a +13.6% accuracy boost on the fine-grained $V^{*}$ benchmark.

**Strengths:**

Excellent Problem Diagnosis: The paper provides a clear and valuable conceptual framework by identifying and naming "mesoscopic bias" and "interpolation bias". This diagnosis of why CLIP fails at high resolutions is a useful contribution to the field.Strong Target-Task Performance: The +13.6% gain on the $V^{*}$ benchmark is extremely significant.

This is a challenging benchmark designed to test the exact fine-grained failures of MLLMs, so this result strongly suggests the method is effective for its intended purpose.Token Efficiency: The quadtree mechanism (QtP) not only prunes tokens but can also improve performance with fewer tokens than the baseline.

This is a highly desirable outcome, demonstrating a successful, content-aware pruning strategy that removes noise/redundancy rather than just information.Low-Cost Intervention (in theory): The core idea of replacing a small part of the vision encoder without retraining the entire MLLM is highly practical and computationally appealing.

**Weaknesses:**

1. The paper's components are not fundamentally new. Quadtrees are a classic data structure, and their application to vision transformers and dynamic tokenization has been explored. Similarly, the token pruning/merging field is already well-established. The idea of replacing static positional embeddings with a dynamic, coordinate-based MLP is also a known technique (e.g., in NeRF). The novelty lies in the combination for this specific MLLM problem, but the components themselves are iterative.

2. The method's impressive gains are not general. The +13.6% boost on $V*$ is met with a performance decrease on MM-Bench (-2.8% for 7B) and stagnation or minor losses on several other benchmarks (e.g., Sci-QA, CV-Bench). This strongly suggests the method is an overspecialization, it trades general VQA capability for fine-grained acuity. The gains do not generalize.Misleading Baselines: The paper highlights outperforming the $S^{2}$ model. However, it also cites SEAL, which achieves a much higher 75.4% on $V^{*}$ overall. While SEAL requires full retraining, this massive 17-point performance gap shows that QLIP is far from the state-of-the-art in solving fine-grained VQA, even if its method (drop-in) is cheaper.

3. "Plug-and-Play" Claim is Misleading: The method is not "plug-and-play." It introduces highly sensitive new hyperparameters, namely the image size and the quadtree pruning threshold $\alpha$. The paper states that the reported results are the "best score from our sweeps". Figure 6 clearly shows a chaotic and non-monotonic relationship between $\alpha$, image size, and performance. A user cannot simply "drop in" QLIP; they must perform an expensive, multi-axis hyperparameter sweep for their specific task to have any hope of replicating the results, which undermines the entire premise of a simple, no-cost intervention."

4. No Retraining" Claim is Misleading: The paper repeatedly claims "no re-training". However, the core MLP interpolation network must be trained. The authors trained it for 11 hours on four NVIDIA L40S GPUs. While this is vastly cheaper than retraining an MLLM, it is not "no retraining." It is a separate, required training step that introduces a dependency on a new dataset (Imagenette) and process.

**Questions:**

1. On Performance Inconsistency: The performance on $V*$ is excellent, but the model gets worse on MM-Bench. Do the authors have a hypothesis for this trade-off? Does the QtP's gradient-based pruning accidentally discard global, "mesoscopic" context that is necessary for general VQA benchmarks but less important for $V^{*}$?

2. On Hyperparameter Sensitivity: Given the extreme sensitivity to $\alpha$ and image size shown in Figure 6, how do you justify the "plug-and-play" claim? What concrete, general-purpose default values would you recommend to a user who cannot afford to run the "sweeps" you performed?On MLP Training: The MLP was trained on Imagenette.

3. How confident are you that this MLP generalizes to image domains outside of natural images (e.g., medical scans, line drawings, or thermal imagery)? Does the "drop-in" claim only apply if the target domain is similar to Imagenet?

4. On Inference Overhead: What is the latency overhead of the quadtree patchification step during inference? Calculating the gradient map and recursively building the tree seems computationally non-trivial compared to a simple uniform grid patch.

---

> ### Author Response · Authors · 2025-12-04
> **Discussion / Rebuttal - Review 1**
>
> We thank the reviewer for their time and their detailed comments on our work. We have been deeply appreciative of the reviewers that were assigned to us in this cycle, especially in light of some of the stories that have been coming out about other ICLR review situations.
>
> The reviewer highlights our excellent problem diagnosis and the significant gains that we obtained on the challenging V* benchmark. They suggest that our methods of avoiding having to retrain the LLM is highly practical.
>
> ### Weaknesses
> 1. We disagree with the assertion of the reviewer that quadtrees’ “application to vision transformers and dynamic tokenization has been explored”. There is a single paper written before ours which uses quadtrees at an embedded token level, but ours is the first study to use quadtrees as the patchification method. After our work was released there has been at least one other author who explored using quadtree patchification at the image level. We also note that to our knowledge our paper is the first application of MLP interpolation to a high-dimensional input space. We chose not to focus on this aspect, but we believe that the demonstration that MLP interpolation techniques continue to work in high-dimensional spaces is also novel.
>
> 2. Our goal was to address the specific case of V*, where the biases we discuss work together to make baseline performance poor, despite the fact that the underlying model is more than capable of answering the questions, as we demonstrate. We address our hypothesis as to why we underperform on MM-Bench in "Questions" below.
>
>     We were hesitant to include the SEAL baseline as their methodology is significantly different than ours. Not only do they retrain their model, but they introduce a search algorithm and set up a visual working memory system. We believe that comparisons between QLIP and SEAL are not apples-to-apples. QLIP demonstrates that the existing model weights and inference pipeline are more than sufficient to dramatically improve fine-grained VQA performance. Finally, we point out that despite all this, QLIP **outperforms** SEAL on POPE, demonstrating the efficacy of our method when compared to much more invasive training interventions.
>
> 3. We note that the non-monotonic behavior noted by the reviewer is a feature, not a bug. The inverted quadratic behavior means that performance **increases** as we decrease token counts, indicating the saliency of our theoretical approach.
>     Finally, we suggest that the reviewer has misunderstood the nature of the sweeps. Given an inference domain, the user needs only to find the optimal \alpha value for their use case. If the inputs are natural images than users should air on the side of using the largest image size that the MLP interpolation network was trained on, corresponding to the best performance in Figure 6.
>
> 4. The MLP network is not re-trained since there are not pre-trained weights which we base the MLP off of.  The advantage of our method is that a lightweight MLP network can be easily trained and the costly and tenuous **re-training** process of a vision encoder and / or LLM is avoided.
>
> ### Questions
> 1. Yes. Our primary hypothesis is that the MM-Bench data is contained in the LLaVA or Vicuna training data. There are some recent works which indicate this may be the case (https://arxiv.org/html/2411.03823v1). We observed that MM-Bench performance is highly sensitive to minor perturbations in the model weights, indicating some form of sensitivity which would not be expected from a fully held-out validation set or benchmark. MM-Bench is also low resolution, which means that QLIP has less advantage when compared to a higher resolution dataset like V*.
>
> 2. We do not believe that Figure 6 shows "extreme sensitivity" to alpha, but rather shows a gradual change in performance as we change \alpha. It is challenging to envision a scenario where someone is running model inference but does not have the compute to run ~2000 user queries through their system. Given that the current paradigm for adapting a model to a user's use case typically involves fine-tuning an off the shelf model, our method represents a monumentally cheaper alternative.
>
> 3. This is a fantastic question. We are not confident that the MLP generalizes to other domains. There is reason to suspect that it will: namely that the training signal from the images mainly serves to align the interpolation. Because the CLIP weights themselves aren't updated, the model has very little capacity to overfit to the training domain. However, there is also a reason to suspect that the MLP won't transfer to other domains: there is inherently **some** alignment of the interpolation to the training domain, and we did not quantify this alignment.
>
> 4. It adds a negligible overhead compared to processing the tokens, and can be heavily optimized using C++ instead of Python to make it hundreds to thousands of times faster.

---

### Meta-Review · Area_Chair_TxbD · 2026-01-08

**Summary:**

The paper identifies limitations of CLIP’s vision encoder (mesoscopic bias and interpolation bias) and proposes a plug-and-play solution, whose core components here are quadtrees and coordinate-based MLPs. The paper receives 6, 4, 6, 6 ratings: Reviewer y52t (6), Reviewer cr9P (4), Reviewer a7Xx (6), Reviewer 5fie (6).

Major concerns

1. Relevance. Reviewer cr9P and Reviewer 5fie question the relevance of the work given that many recent methods can handle arbitrary resolutions.

2. Novelty: Reviewer y52t finds the components to be “iterative” (though the combination is not).

3. Experimental results: Reviewer a7Xx and Reviewer 5fie would like to see stronger/more recent models other than LLaVa-1.5 (7B and 13B).

4. Limitations in other tasks. Reviewer cr9P raises a concern regarding CLS bias and whether this approach would perform well on dense prediction tasks (segmentation, object detection). Reviewer a7Xx wants to see results on visual grounding tasks.

**Reviewer Concerns:**

1. [Somewhat resolved] CLIP is still widely used. Also see item 3 below.

2. [Somewhat resolved] The authors disagree using quadtrees related work as an example.

3. [Not resolved, Future work] Models like InternVL-3.5 or QwenVL2.5 use complex vision encoding schemes that are difficult to adapt to quadtree patchification.

4. [Not resolved] The authors do not provide response or other results. The authors admit “The strength of our method is also a weakness in certain contexts, although we think that the reviewer’s statement can be bolstered by some nuance.
Our method is not intended to be good at segmentation tasks per-se…”

Other non-major concerns

[Resolved] Claims on simplicity (hyperparameter-sensitivity, training-free, plug-and-play) : Reviewer y52t

[Resolved] Generality; Slight regression on non fine-grained tasks: Reviewer y52t. The authors provide a hypothesis as to why they underperform SEAL on MM-Bench. (Not only do they retrain their model, but they introduce a search algorithm and set up a visual working memory system).

**Reviewer Scores:**

Reviewer y52t (6?). Keep or increase.

Reviewer cr9P (4). Keep.

Reviewer a7Xx (6). Keep.

Reviewer 5fie (6). Keep.

---

### Decision · Program_Chairs · 2026-01-26

Accept (Poster)